# Genomic analyses of rice bean landraces reveal adaptation and yield related loci to accelerate breeding

Jiantao Guan[1,2,3,10], Jintao Zhang[1,4,10], Dan Gong[1,4,10], Zhengquan Zhang[2], Yang Yu [2], Gaoling Luo[5], Prakit Somta[6], Zheng Hu[1], Suhua Wang[1], Xingxing Yuan[7], Yaowen Zhang[8], Yanlan Wang[9], Yanhua Chen[5], Kularb Laosatit[6], Xin Chen[7], Honglin Chen[1], Aihua Sha[4], Xuzhen Cheng[1], Hua Xie [2] ⬚ & Lixia Wang [1] ⬚

Rice bean (*Vigna umbellata*) is an underexploited domesticated legume crop consumed for dietary protein in Asia, yet little is known about the genetic diversity of this species. Here, we present a high-quality reference genome for a rice bean landrace (FF25) built using PacBio long-read data and a Hi-C chromatin interaction map, and assess the phylogenetic position and speciation time of rice bean within the *Vigna* genus. We sequence 440 landraces (two core collections), and GWAS based on data for growth sites at three widely divergent latitudes reveal loci associated with flowering and yield. Loci harboring orthologs of *FUL* (*FRUITFULL*), *FT* (*FLOWERING LOCUS T*), and *PRR3* (*PSEUDO-RESPONSE REGULATOR 3*) contribute to the adaptation of rice bean from its low latitude center of origin towards higher latitudes, and the landraces which pyramid early-flowering alleles for these loci display maximally short flowering times. We also demonstrate that copy-number-variation for *VumCYP78A6* can regulate seed-yield traits. Intriguingly, 32 landraces collected from a mountainous region in South-Central China harbor a recently acquired InDel in *TFL1* (*TERMINAL FLOWER1*) affecting stem determinacy; these materials also have exceptionally high values for multiple human-desired traits and could therefore substantially advance breeding efforts to improve rice bean.

The genus *Vigna* is a pan-tropical genus in the family Fabaceae, comprising more than 100 wild species and 10 domesticated species such as cowpea (*Vigna unguiculata*), mung bean (*V. radiata*), and rice bean (*V. umbellata*)[1]. As one of the representative species in the genus *Vigna*, the rice bean is a multipurpose legume and is widely cultivated in South, Southeast, and East Asia[2]. The seeds of rice beans have been consumed for thousands of years as a good source of dietary protein and micronutrients, and these are used as a diuretic in traditional medicine practices[3,4]. Rice bean has also been widely used as a donor parent for interspecific hybridization with other species in the genus *Vigna*[5–7] due to its notable agronomic characteristics including high grain yield and large biomass potential[2,8], as well as strong resistance to pests[9–14], diseases[15], drought[2,16,17], water logging[18], and capacity to grow in poor fertility soils[19]. Thus, as the continually growing population and exacerbated climate changes, rice bean has received increased attention in recent years and has been proposed as one of the potential future smart foods to help to fight hunger and malnutrition in Asia[2,18,20]. However, the lack of a high-quality reference genome for rice beans has hindered the exploration of the genetic basis of these excellent agronomic characteristics and its further genetic improvement.

Current thinking holds that the rice bean originated and was domesticated in tropical regions of South & Southeast Asia, after

A full list of affiliations appears at the end of the paper. ⬚ e-mail: xiehua@baafs.net.cn; wanglixia03@caas.cn

which it spread to higher latitude regions including China, Japan, and Korea[2,21,22]. There are many rice bean landraces that have, through long-term human and natural selection, become locally adapted to diverse environments. However, as a short-day plant, the yield potential and agricultural utility of rice beans can be strongly affected by photoperiod and temperature conditions[2,23,24]. Moreover, few cultivated rice bean varieties have a determinate stem growth habit that influences the potential grain yield and is also required to support mechanical harvest[6,25]. Landraces have been demonstrated as useful resources for the improvement of diverse crop species[26], and there are presently two rice bean core collections, one comprising mainly landraces from South & Southeast Asia and the other with a preponderance of Chinese rice bean landraces[22,27,28]. Thus, there are rich germplasm panels available representing the high diversity and broad adaptation of rice beans to both tropical and temperate environments.

Previous studies have reported several QTLs for adaptation and yield component-related traits using linkage mapping based on biparental populations in rice bean[11,29]. However, the resolution and sensitivity have been limited by the small number of markers and genetic recombination, thus making it difficult to reveal the genetic mechanism of these traits and/or to develop breeding markers[30,31]. Genome-wide association studies (GWAS) have been successfully applied in crops for the efficient identification of favorable alleles/haplotypes or causal variants/genes underlying complex traits as this strategy could simultaneously detect many natural allelic variations using a diverse germplasm panel[32,33].

Here, we present a high-quality reference genome assembly of rice beans based on the integration of Illumina short-reads, PacBio long-reads, and Hi-C sequencing data. We also construct a genome variation map based on sequencing of 440 diverse rice bean landraces covering two core collections. Subsequent population genomic analyses support the previously proposed origin of rice bean in South & Southeast Asia and revealed genetic bottlenecks that occurred along the northward dispersal of rice bean. GWAS based on phenotypic data for a germplasm diversity panel grown at three sites with widely divergent latitudes helps decipher the genetic basis of traits including flowering time, seed yield, and stem determinacy. Our study also identifies candidate genes and landraces with strong potential as elite germplasm lines that could be used to generate excellent varieties that simultaneously display geographically suitable flowering times, stem determinacy to support mechanized cultivation, and high yields of rice beans.

## Results

### Sequencing and assembly of a reference genome for rice bean

The rice bean landrace FF25—which has red seeds, an erect habit, and wide environmental adaptability—was selected for genome sequencing and de novo assembly of a rice bean reference genome (Fig. 1a). We integrated three sequencing technologies: PacBio single molecule real-time (SMRT) long-read sequencing, Illumina short-read sequencing, and chromosome conformation capture sequencing data (Hi-C) (Supplementary Table 1). The estimated genome size of the FF25 genome was ~525.60 Mb based on 17-kmer depth distribution using

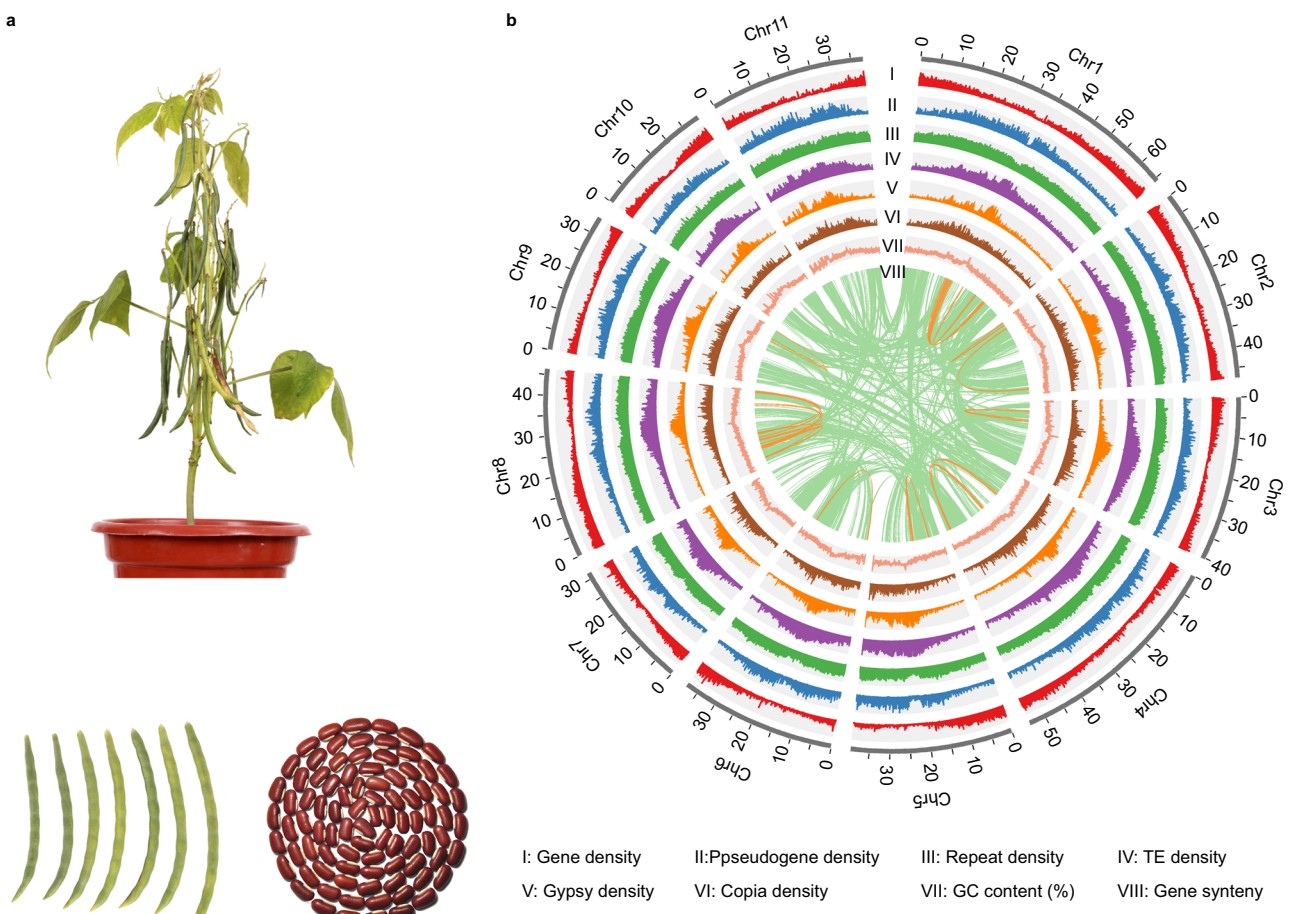

I: Gene density  II: Ppseudogene density  III: Repeat density  IV: TE density
V: Gypsy density  VI: Copia density  VII: GC content (%)  VIII: Gene synteny

**Fig. 1 | FF25 genome assembly. a** FF25 plant (Top); FF25 pod (Bottom left); FF25 seeds (Bottom right). **b** Genomic features of the FF25 reference genome. The outer gray track represents the chromosomes of the FF25 genome assembly (with units in Mb). The densities of features were calculated based on 100 kb window size, with a step size of 10 kb. The inner green and orange links represent the intra- and inter- chromosomal collinear genes, respectively. Photograph credit: LXW (**a**).

**Table 1 | Summary statistics for the rice bean genome assembly**

| Genomic feature | Value |
| --- | --- |
| Total assembly size (Mb/%) | 475.64/90.49% |
| Number of contigs | 351 |
| Largest contigs (Mb) | 32.05 |
| Contig N50 (Mb) | 18.26 |
| Sequences anchored to chromosomes (Mb) | 465.19/97.80% |
| Genomic GC content (%) | 34.21 |
| Genome Complete BUSCOs[a] (%) | 97.3 |
| Protein Complete BUSCOs[a] (%) | 96.9 |
| LTR assembly index, LAI | 20.30 |
| Repetitive sequences (%) | 57.19 |
| Protein-coding genes | 26,736 |
| Mean gene length (bp) | 3,602 |
| Mean coding sequences/exon/intron length (bp) | 1232/238/570 |

[a]Analysis based on comparisons with the eudicotyledons_odb10 database.

Illumina short-reads (~106.65×; Supplementary Fig. 1). The PacBio reads (~300.54×) were used to assemble the contigs using Canu v1.9[34] and the highly efficient repeat assembly (HERA) algorithm[35], which resulted in a 475.64 Mb genome (90.49% of the estimated size) containing 351 contigs, with an N50 of 18.26 Mb (Table 1), thus representing highest quality genome among species of the *Vigna* genus[36–40]. To assign the contigs to different chromosomes, 66 contigs (~465.19 Mb, 97.80% of the original assembly) were anchored to eleven pseudo-chromosomes based on a Hi-C interaction map (Table 1; Supplementary Fig. 2).

We used multiple methods to evaluate the quality of the assembled genome. The mapping and coverage rates of the Illumina short-read data were 99.67% and 99.33%, respectively. We further performed benchmarking universal single-copy orthologs (BUSCO) analysis[41] based on the eudicotyledons_odb10 dataset, and the result showed that 97.3% of the BUSCO sequences were completely present in the genome assembly, while 0.5% and 2.2% were partially present or missing, respectively (Table 1; Supplementary Table 2). The genome assembly had a high LTR Assembly Index (LAI) score (20.30) (Supplementary Table 2; Supplementary Fig. 3), reaching the level of a gold standard genome assembly according to previously proposed criteria[42]. All of these lines of evidence indicate that our de novo FF25 genome assembly is of high quality.

We used an integrated strategy including evidence-based methods and ab initio gene prediction to annotate the protein-coding gene content of the FF25 genome assembly. A final set of 26,736 protein-coding genes was predicted, of which 26,430 genes (~98.86%) could be assigned to eleven pseudomolecules (Supplementary Table 3). Of these genes, the average lengths of coding sequences, exons, and introns were 1232 base pairs (bp), 238, and 570 bp, respectively (Table 1). The average gene density was one gene per 17.79 Kb, and the genes were unevenly distributed, being more abundant towards the chromosomal ends (Fig. 1b). We also specifically concatenated 2202 transcription factor genes, 9635 pseudogenes, and 3318 noncoding RNA genes comprising 764 transfer RNA genes, 558 ribosomal RNA genes, 714 small nucleolar RNA genes, and 1282 microRNA genes (Fig. 1b; Supplementary Table 4).

Of these predicted protein-coding genes, we found that 96.90% of the BUSCO sequences were completely present (Table 1; Supplementary Table 2). Moreover, the tissue-specific RNA-Seq data confirmed that 85.86% of the predicted protein-coding genes were expressed (FPKM > 1) in at least one of the 6 examined tissues (Supplementary Table 5). And 97.48% of the protein-coding genes were assigned a functional annotation based on five public databases (Supplementary

Table 6). These evaluations collectively support the high accuracy and completeness of our rice bean genome assembly and annotation.

## Phylogenetic position and comparative genomics analyses

To explore the genome evolution of rice bean, genes from the five *Vigna* species (*Vigna stipulacea*, *V. radiata*, *V. angularis*, *V. umbellata*, and *V. unguiculata*), four other legumes (*Phaseolus vulgaris*, *Glycine max*, *Lotus japonicus*, and *Arachis duranensis*), five other eudicots (*Arabidopsis thaliana*, *Citrus sinensis*, *Populus trichocarpa*, *Vitis vinifera*, and *Solanum lycopersicum*), as well as one monocot (*Oryza sativa*) were clustered into 20,736 gene families. Of these, 334 single-copy gene families were used to construct a maximum-likelihood phylogenetic tree (Fig. 2a). This indicated that rice bean is a sister species to adzuki bean (*V. angularis*); they apparently diverged about 1.75 million years ago (MYA), findings in accord with a previous study based on transcriptome data[37].

This view was also supported by a gene synteny analysis between rice bean and its closely related species in the *Vigna* genus based on protein sequences using the MCScanX program[43], which revealed that (as expected) rice bean had higher conservation with *V. angularis* in terms of gene structure and order as compared to other *Vigna* species (Supplementary Fig. 4; Supplementary Table 7). Based on the tree, we found that 230 rice bean gene families (comprising 1396 genes) exhibited significant expansions ($P < 0.01$) relative to the MRCA (most recent common ancestor) of rice bean and adzuki bean (Supplementary Data 1). KEGG pathway analysis indicated that these expanded genes were significantly enriched for metabolism pathways such as the phenylpropanoid, sesquiterpenoid, and triterpenoid biosynthesis ($P < 0.05$, Fisher's exact test; Supplementary Fig. 5).

Whole-genome duplication (WGD) provides additional genetic material that can be subsequently subjected to divergence, subfunctionalization, and neofunctionalization[44,45]. To investigate WGD events in rice bean, we identified 332 syntenic blocks within its genome (including 8052 homologous genes accounting for ~30.12% of all genes) (Fig. 1b) and estimated synonymous nucleotide substitutions at synonymous sites (Ks) for homologs. The Ks distribution of collinear gene pairs indicated no recent WGD in rice beans; we also observed the expected signals for the ECH event (eudicot-common hexaploidy; Ks = 1.72) and the LCT (legume-common tetraploid; Ks = 0.64) event (Fig. 2b). We estimated the relative time of evolutionary divergence between rice bean and closely related *Vigna* species using the Ks distributions of orthologs based on the known evolutionary time (~13 MYA) of the SST (soybean-specific tetraploid) event in soybean[37,46]. Similar to the very recent speciation time estimated from the maximum-likelihood phylogenetic tree (Fig. 2a), the Ks distribution of rice bean and adzuki bean also showed the smallest peak value at 0.019 (Fig. 2b), corresponding to a divergence time of 1.72 MYA.

Beyond comparisons of orthologous genes, we annotated the repetitive content in the rice bean genome using an integrated pipeline, including de novo repeat identification and homology search methods (see the "Methods" section). We identified that 38.40% of the rice bean genome comprises transposable elements (TEs; Supplementary Data 2). Among the distinct classes of TEs, LTR elements including *Gypsy* and *Copia* elements were the predominant classes; and compared to *Copia* elements (10.41%), *Gypsy* elements (19.85%) occupied relatively larger proportions of genomic sequence in rice bean, which is consistent with earlier reports about other *Vigna* species[37,40]. In addition, we identified full-length LTR elements and performed an insert time analysis for rice bean as well as other four additional *Vigna* species with sequenced genome assemblies. Excepting *Vigna radiata*, more than half of the LTR elements in the other four examined *Vigna* species proliferated at 0 – 0.5 MYA, suggesting that the amplification of LTR elements has largely occurred after speciation (Fig. 2c).

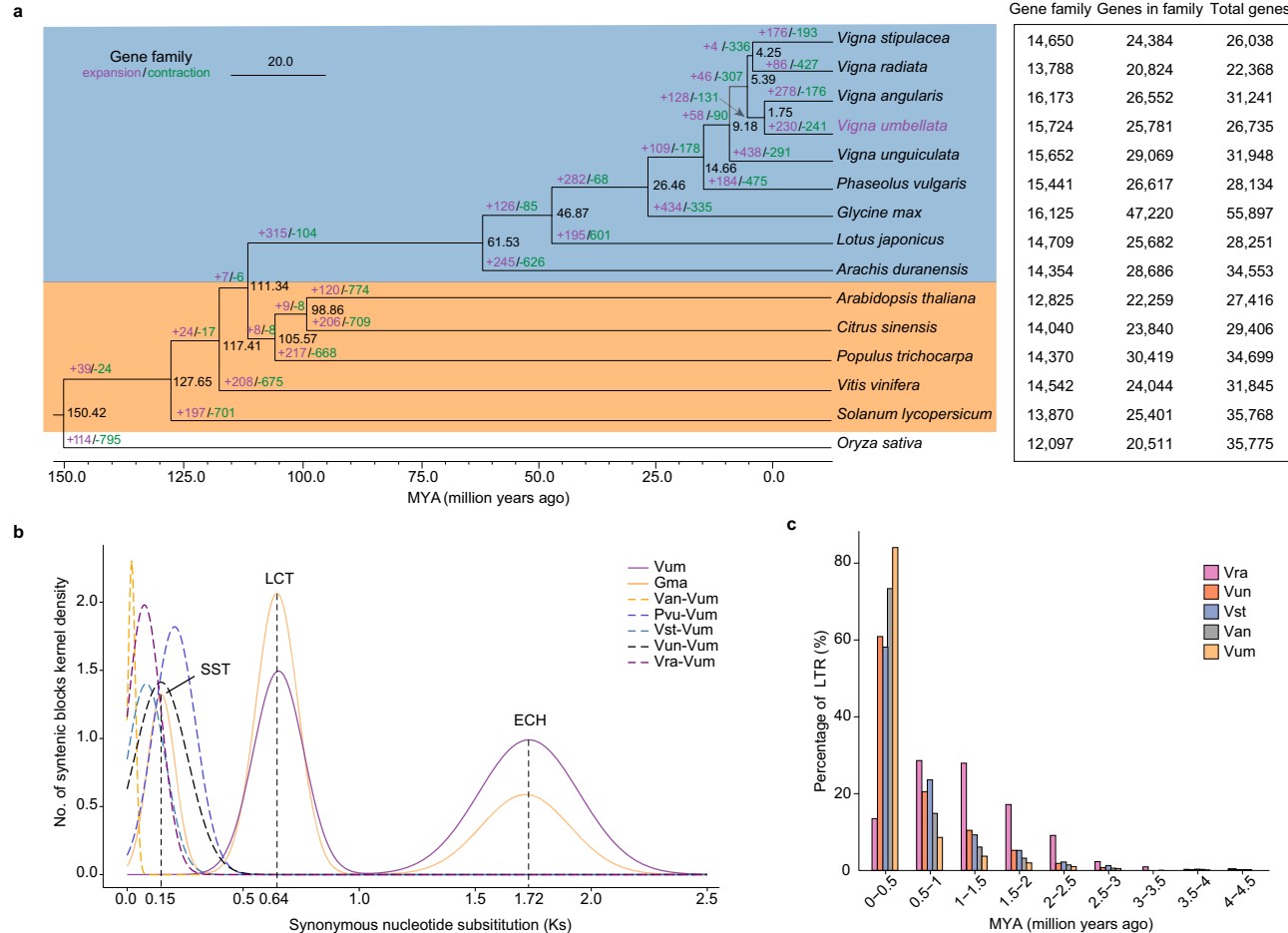

| Gene family | Genes in family | Total genes |
|---|---|---|
| 14,650 | 24,384 | 26,038 |
| 13,788 | 20,824 | 22,368 |
| 16,173 | 26,552 | 31,241 |
| 15,724 | 25,781 | 26,735 |
| 15,652 | 29,069 | 31,948 |
| 15,441 | 26,617 | 28,134 |
| 16,125 | 47,220 | 55,897 |
| 14,709 | 25,682 | 28,251 |
| 14,354 | 28,686 | 34,553 |
| 12,825 | 22,259 | 27,416 |
| 14,040 | 23,840 | 29,406 |
| 14,370 | 30,419 | 34,699 |
| 14,542 | 24,044 | 31,845 |
| 13,870 | 25,401 | 35,768 |
| 12,097 | 20,511 | 35,775 |

**Fig. 2 | Phylogenetic position and comparative genomics analyses. a** Genome evolution and gene family characteristics of *Vigna umbellata* (rice bean) and 13 other dicot species using the monocot plant *Oryza sativa* (rice) as an out-group. This tree was generated using 334 single-copy ortholog families. Black numerical values beside each node show the estimated divergence time of each node (MYA, million years ago) in the phylogenetic tree shown on the left. Blue and orange backgrounds represent *Leguminosae* and non-*Leguminosae* species, respectively. The number of gene families, genes in the family, and the total number of genes are

shown on the right for each species. **b** Density distribution of synonymous nucleotide substitution levels (Ks) of syntenic orthologous (solid curves) and paralogous genes (dashed curves). Vum: *Vigna umbellata*; Gma: *Glycine max*; Van: *V. angularis*; Pvu: *Phaseolus vulgaris*; Vst: *V. stipulacea*; Vun: *V. unguiculata*; Vra: *V. radiata*. **c** Insertion bursts of full-length LTR elements in the genomes of *V. umbellata* and other four *Vigna* species. Source data are provided as a Source Data file.

## Population structure and genetic divergence of rice bean landraces

We performed whole genome re-sequencing for a total of 440 rice bean landraces from various geographic regions, including the land-races in the Asia core collection (73) and Chinese core collection (230) using Illumina sequencing technology (Fig. 3a), ultimately generating 5.32 Tb of high-quality sequencing data, with an average depth of ~24.91× and an average mapping rate of 99.12% based on the newly assembled reference genome (Supplementary Data 3). A final set of 10,525,548 high-quality single-nucleotide polymorphisms (SNPs) and 2,743,289 small insertions and deletions (InDels) were identified. We found 5690 SNPs (0.054%) that caused start codon changes, pre-mature stop codons, or elongated transcripts, while 15,530 InDels (0.57%) lead to frameshift mutations (Supplementary Table 8), pro-portions similar to other species likely soybean[47], cucumber[48], and watermelon[49].

To infer the population structure, we constructed an SNP-based neighbor-joining (NJ) phylogenetic tree and divided the 440 landraces into three geographical groups: landraces from South & Southeast Asia (SSA), South China (SC; coastline of South China to the Yangtze River), and North China (NC; Yangtze River to North China) (Fig. 3a, b; Sup-plementary Data 3). This classification was supported by a principal

component analysis (Fig. 3c) as well as a model-based clustering ana-lysis (K = 4) conducted using STRUCTURE[50] (Fig. 3b). Notably, the landraces collected from other geographical regions (Japan, Korea, Europe, and America) were spread amongst the SC and NC groups, indicating their close genetic relationship with Chinese landraces or their probable introduction from China[2]. We excluded these landraces from the SC and NC groups in our further analyses.

To investigate genetic diversity and divergence among the three geographical groups, we calculated the nucleotide diversity ($\pi$) for each group and conducted a pairwise analysis of genetic distances (Fixation index values, $F_{ST}$). The SSA group showed the highest nucleotide diversity ($1.08 \times 10^{-3}$), consistent with the previous results using SSR markers[22] and further supporting the hypothesis that rice beans originated from South & Southeast Asia[2,22]. Compared with the SSA group, gradually decreased nucleotide diversity was observed in the SC group ($0.78 \times 10^{-3}$) and then the NC group ($0.43 \times 10^{-3}$), indi-cating that sequential bottlenecks ($\pi_{SSA}/\pi_{SC} = 1.38$; $\pi_{SC}/\pi_{NC} = 1.81$) occurred during the northward dispersal of rice bean from the origin center (Fig. 3d). When compared with the SSA group, the $F_{ST}$ value of the SC group was 0.16, whereas it became higher (0.29) for the NC group, indicating enlarged population differentiation during the northward dispersal (Fig. 3d).

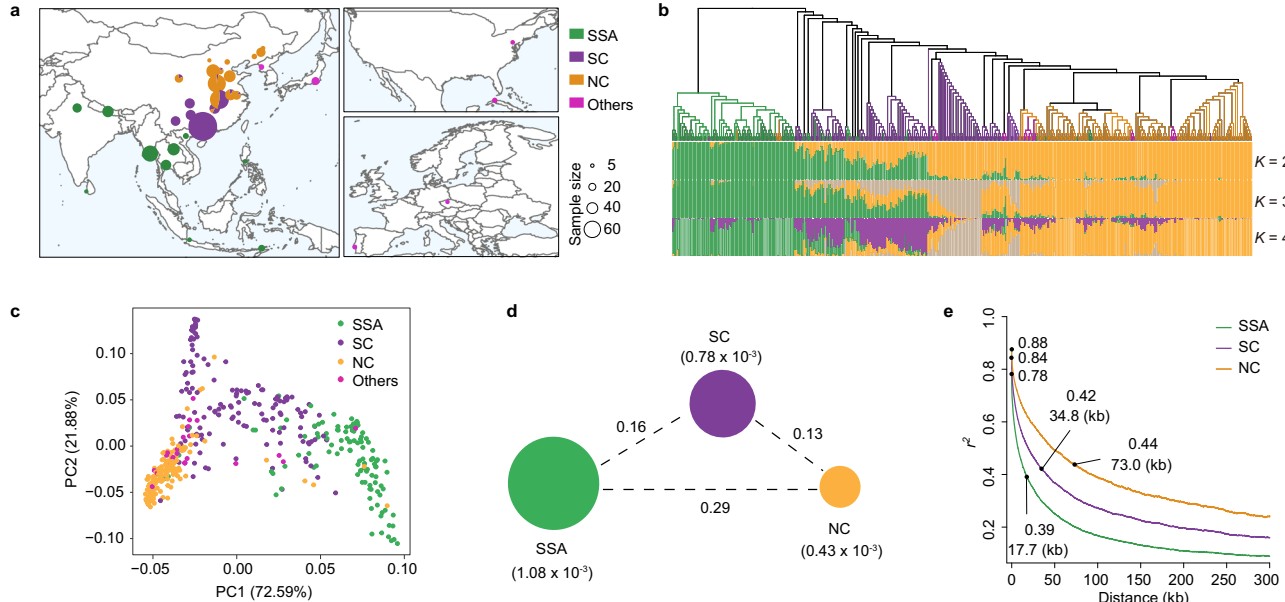

**Fig. 3 | Population structure and genetic divergence of rice bean landraces.**
**a** The geographic distributions of 440 rice bean landraces. SSA South & Southeast Asia, SC South China, NC North China. The size and color of each pie chart represent the sample size in a specific geographic location. The map was created using the map_data() function in the R package ggplot2. **b** Phylogenetic tree and model-based clustering ($K = 2–4$) of 440 sequenced landraces. **c** Scores plot from a principal component analysis, supporting the division of the landraces into three geographical groups (SSA, SC, and NC). **d** Summary of nucleotide diversity ($\pi$) and population divergence ($F_{ST}$) across the three geographical groups. Values in parentheses represent measures of nucleotide diversity of each group, and values between pairs indicate population divergence. **e** Decay of linkage disequilibrium (LD), measured by $r^2$, in the three geographical groups. The upper and lower black dots with numerical values in the lines represented maximum and median values of the $r^2$ and the corresponding physical distances. Source data are provided as a Source Data file.

We further examined linkage disequilibrium (LD) using the measure ($r^2$)[51] between pairwise SNP loci in SSA, SC, and NC groups. For the SSA group, the decay of LD with physical distance (i.e., a drop to half of its maximum value) between SNPs occurred at only ~7.7 kb ($r^2 = 0.39$), whereas it increased to ~34.8 kb ($r^2 = 0.42$) in the SC group and to ~73.0 kb ($r^2 = 0.44$) in the NC group (Fig. 3e); these trends are in accord with the observed gradual reduction in genetic diversity in the SC and NC groups. The LD of rice bean landraces was similar to those of outcrossing species such as maize (30 kb)[52] but shorter than those of inbreeding crops like soybean (83 kb)[53], rice (123 kb and 167 kb in *indica* and *japonica*, respectively)[54], and foxtail millet (~100 kb)[55]. This finding is consistent with a previous report that rice bean has a fairly high outcrossing rate[22]. Notably, the relatively rapid LD decay in the rice bean landraces may be useful for enhancement of resolution power of association studies to map a narrow candidate QTL interval[56].

We searched for putatively selective regions with outliers (top 5%) of $F_{ST}$ over 20-kb windows for the three comparisons (SSA vs. SC, SSA vs. NC, and SC vs. NC). We detected 473, 512, and 444 outlier regions for these three comparisons, respectively occupying 5.67% (26.95 Mb), 5.92% (28.15 Mb), and 5.59% (26.57 Mb) of the genome and including 1894, 1950, and 1296 protein-coding genes (Supplementary Data 4). A MapMan analysis of all the selected genes indicated that these genes were significantly enriched for annotations related to biological processes such as "phytohormone action", "nutrient uptake", and "circadian clock system" (Supplementary Fig. 6). Notably, among the genes related to "circadian clock system", we found four orthologs of reported flowering time genes in *A. thaliana* using FLOweRing Interactive Database (FLOR-ID)[57], including *TOC1*[58], *PRR3*[59], and two *LHY1*[60] genes between the SSC and SC groups, of which one *LHY1*[60] apparently also underwent selection between the SSC and NC groups (Supplementary Fig. 7). These flowering time genes could plausibly have contributed to the adaptation of rice bean landraces to different latitudes.

## The genetic architecture underlying control of flowering at different latitudes

We observed flowering time variation across 440 landraces as grown at three sites with widely divergent latitudes: 22–106 days in Sanya (18°N) in 2020 and 2021, 25–122 days in Nanning (22°N) in 2020 and 2021, and 38–104 days in Beijing (40°N; where some landraces did not bloom before the first frost in the autumn of 2020 and 2021). To explore the genetic basis of the flowering time for rice beans, we performed GWAS for the flowering phenotype data measured in both years at the three sites, which revealed distinct association signals for the different locations (Fig. 4a–c). The repeatedly detected major signal from Sanya was an intergenic region (Chr11: 6,142,933–6,162,249) that was only ~5 kb away from a MADS-box gene that is the closest rice bean homolog (*Vum_11G00418*) of Arabidopsis *FRUITFUL* (*FUL*) (Fig. 4a; Supplementary Figs. 8, 9; Supplementary Table 9), a gene known to control flowering time and reproductive transition[61].

This GWAS signal explained up to 7.04–14.86% of the flowering time variation across two years (Supplementary Data 5). All the significantly associated SNPs and InDels in this GWAS signal were located in its upstream region (>5 kb) (Supplementary Fig. 10), suggesting that these polymorphisms could influence *FUL* expression to control flowering time. This was further supported by the observation that the expression level of *FUL* (in newly expanded leaves in a panel of 16 diverse rice bean landraces) was strongly negatively correlated ($R = -0.69$, $P = 2.95 \times 10^{-3}$) with flowering time (Supplementary Fig. 11).

For the Nanning site, the repeatedly detected major signal (Chr4: 35,931,101–35,996,258) had a PVE (phenotypic variation explained) value of 6.07–8.23% (Supplementary Fig. 8; Supplementary Data 5; Supplementary Table 9). And the most likely candidate among the five protein-coding genes in this region is a *FLOWERING LOCUS T* (*FT*; *Vum_04G01668*) ortholog (Fig. 4b; Supplementary Fig. 12); in many species, *FT* genes function as integrators of diverse signals for controlling of flowering time[62]. We found two significantly associated SNPs around (<2 kb) and within *FT* gene; one SNP

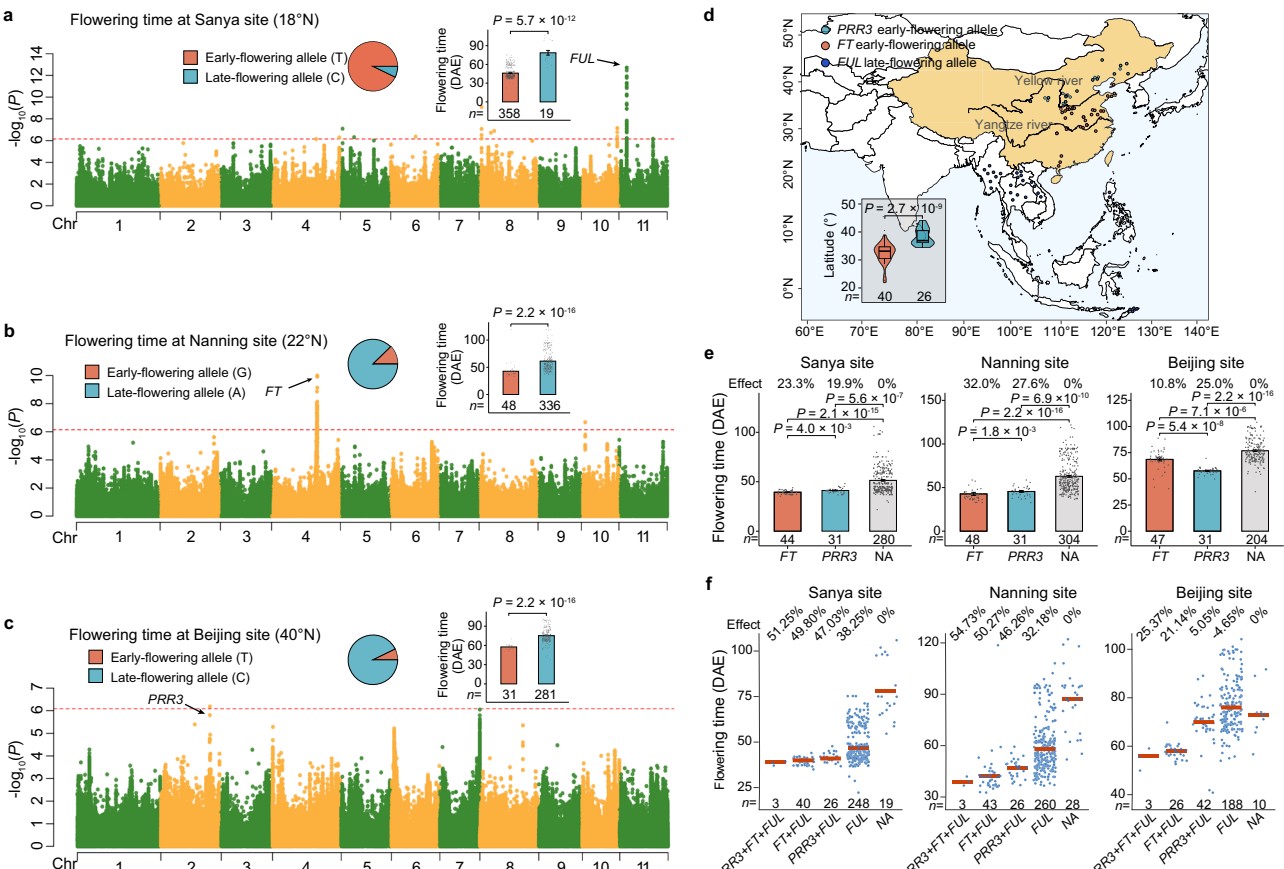

**Fig. 4 | The genetic architecture underlying flowering time control from low to high latitudes. a–c** Manhattan plots of GWAS for flowering-time data measured in Sanya (18°N) in 2021 (**a**), Nanning (22° N) in 2020 (**b**), and Beijing (40°N) in 2021 (**c**). Red horizontal dashed line indicated the Bonferroni-corrected significance thresholds of GWAS ($\alpha = 1$). Pie charts represented allelic frequencies of the major associated loci. The bar plots display the flowering time of landraces carrying each allele of the identified major loci. DAE, days after emergence. The number (n) of landraces carrying each allele is shown below. **d** The geographical distributions of landraces carrying the early-flowering alleles of *FT* and *PRR3*, and the late-flowering allele of *FUL*, respectively. The map was created using the map_data() function in the R package ggplot2. The violin plot showed significant differences in latitudes between landraces carrying the early-flowering allele of *FT* (n = 40) and *PRR3* (n = 26), respectively. In the box plots, central line: median values; bounds of the box: 25th and 75th percentiles; whiskers: 1.5*IQR (IQR: the interquartile range

between the 25th and 75th percentile). **e** The bar plots show the flowering shortening effects of the early-flowering alleles of *FT* and *PRR3* at each of the three measurement sites. NA indicates landraces carrying neither of these two early-flowering alleles. The number (n) of landraces carrying each allele at each of the three measurement sites is shown below. **f** The dot plots show the flowering time shortening effects of early-flowering allelic combinations at each of the three measurement sites. Blue dots represent the landraces categorized according to all of the different allelic combinations found in the 440 sequenced landraces. Red lines indicate the average value of each category. NA indicates landraces carrying no early-flowering alleles. The number of landraces for each category is shown below. The significance was tested with two-sided Wilcoxon tests in (**a**–**e**). The data in **a**–**c** and **e** are shown as mean ± SE, and the error bars represent SE. Source data are provided as a Source Data file.

(Chr4:35,950,445) was located upstream (<200 bp) of the transcription start site and another was located in the first intron (Chr4:35,951,311) (Supplementary Fig. 13).

For the Beijing data, we repeatedly detected a peak SNP in the *PSEUDO-RESPONSE REGULATOR 3* (*PRR3*) gene (*Vum_02G01965*) at Chr2: 38,647,190 (7.30–17.30% of PVE), encoding a nonsynonymous variant (S–F) in the third CDS consisting of the functional PR (pseudo-receiver) domain (Fig. 4c; Supplementary Data 5; Supplementary Figs. 8 and 14; Supplementary Table 9). *PRR3* is an ortholog of the known soybean circadian clock gene *GmTof12/GmPRR3b* that has been previously shown to function as a major flowering time regulatory gene and has been linked to the expansion of soybean into higher latitudes[63,64]. Notably, a similar effect from a single amino acid change (S–L) on flowering time has also been reported for the *GmPRR3b* gene in soybean[63].

We next explored the potential flowering-time-related impacts of the *FUL*, *FT*, and *PRR3* orthologs in rice beans by classifying the landraces according to their alleles at these three loci. There were two

alleles for *FUL* in the collection, and at the Sanya site, the set of 28 landraces carrying the minor allele (6.73%) displayed significantly (P < 0.001) later flowering time (-33 days delayed, a 70.72% increase) than the set of landraces carrying the major allele (Fig. 4a). Note that all of the landraces carrying the late-flowering *FUL* allele were initially collected from low latitude regions (South & Southeast Asia; Fig. 4d). We also found these landraces carrying the late-flowering *FUL* allele also exhibited a significantly higher number of branches than other landraces carrying the early-flowering *FUL* allele (Supplementary Fig. 15), suggesting the probable effect of high yield potential from the late-flowering *FUL* allele.

In contrast, landraces carrying the minor alleles for *FT* (12.31%) and for *PRR3* (7.24%) displayed earlier flowering times, with the average flowering time for the landraces carrying the early-flowering *FT* allele -19 days earlier (a 30.29% reduction) and 18 days earlier (23.36%) for the landraces carrying the *PRR3* minor allele (Fig. 4b, c). There were notable geographical differences among the landraces carrying the early-flowering alleles of *FT* and *PRR3* genes: for *FT* there was a clear

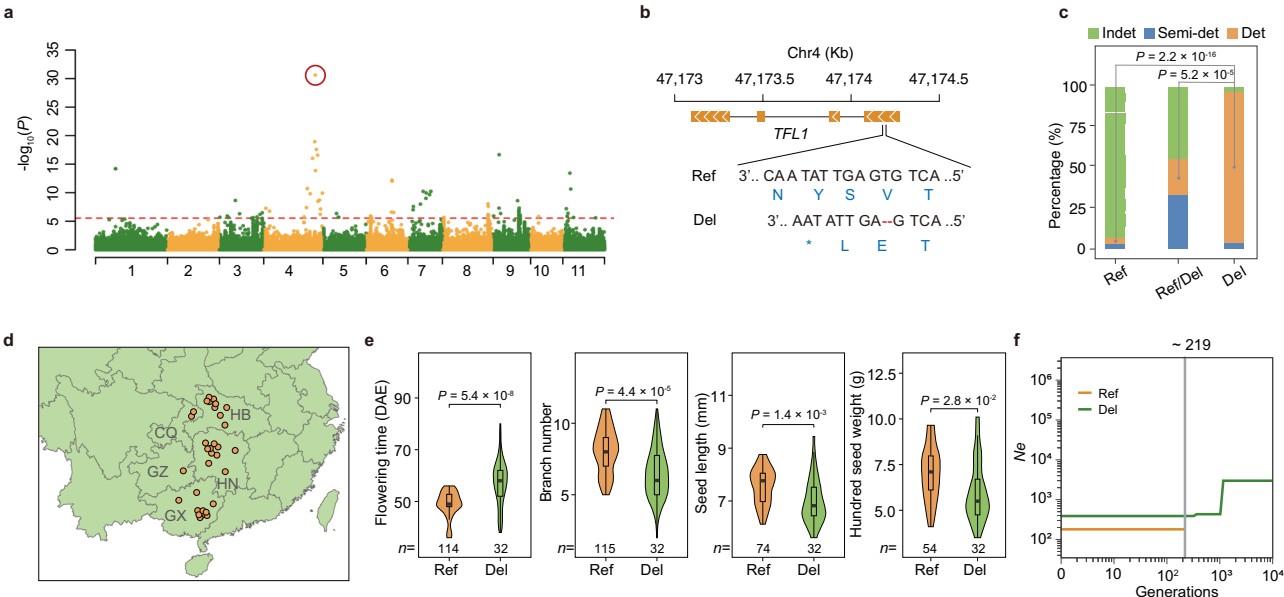

**Fig. 5 | The molecular basis and selection history of stem determinacy in cultivated rice bean. a** InDel-based GWAS result from the analysis of data for stem determinacy measured at the Nanning site in 2020. The peak InDel is indicated by the red circle. The red horizontal dashed line indicated the Bonferroni-corrected significance thresholds of GWAS ($\alpha = 1$). **b** A 2-bp causative deletion (the peak InDel) introduced a premature termination codon in the first exon of the *TFL1* gene. Ref reference, Del deletion. **c** The frequency distributions of three types of stem growth habit (indeterminate (Indet), semi-determinate (Semi-det), and determinate (Det)) among three groups comprising landraces carrying the homologous reference alleles (designated as Ref), heterozygous (Ref/Del) or homologous mutation (2-bp deletion) alleles (Del), respectively. Two-sided Fisher's exact tests were used to assess the significance of the differences in the proportion of the determinate type of stem growth habit between landraces carrying Ref and Del alleles and between landraces carrying Ref/Del and Del alleles. **d** The geographical distributions of the 32 landraces carrying Del alleles from Southern-Central China. The map was created using the map_data() function in the R package ggplot2. HB Hubei province, HN Hunan province, CQ Chongqing province, GZ Guizhou province, GX Guangxi province. **e** There was a significant improvement for the 32 landraces carrying Del alleles compared with the landraces carrying Ref alleles in the SC group for multiple human-desired traits including flowering time (DAE, days after emergence), branch number, seed length, and HGW (hundred seed weight). Significance was tested with two-sided Wilcoxon tests. In the box plots, central line: median values; bounds of the box: 25th and 75th percentiles; whiskers: 1.5*IQR (IQR: the interquartile range between the 25th and 75th percentile). **f** Divergence time of the 32 landraces carrying Del alleles with the landraces carrying Ref alleles in the SC group, inferred using the SMC++ program[66], under a mutation rate $\mu = 1.5 \times 10^{-8}$ per site per generation[140], and a generation time of one year. Source data are provided as a Source Data file.

trend for collection from the region between the Yangtze and Yellow rivers, whereas the landraces harboring the early-flowering *PRR3* allele tended to be from higher latitude regions north of the Yellow River (including Northwest and Northeast China) (Fig. 4d). We also inferred the model of inheritance for these alleles and found that the best models for *FUL*, *FT*, and *PRR3* loci were additive, dominant, and additive, respectively (Supplementary Table 10; see the "Methods" section).

Beyond suggesting that early-flowering alleles for both of these loci have contributed to the adaptation of rice beans to higher latitudes (relative to the tropical origin center), these results indicate potential discrete impacts of the two loci that are sensitive to conditions found in different latitudinal ranges. Offering support for this idea, analysis of phenotype data from the geographically distinct test site revealed differential impacts from the two alleles of interest at the *FT* and *PPR3* loci. That is, at the northernmost site of our study (Beijing), the extent of the flowering time shortening effect was significantly larger among the set of landraces carrying the relevant *PRR3* allele as compared to the set of landraces carrying the relevant *FT* allele (Fig. 4e). Importantly, this trend was reversed at the other two (more southerly) sites: at both Nanning and Sanya, the set of landraces with the early-flowering *FT* allele had the shorter flowering times (Fig. 4e).

We also evaluated the pyramiding effects of the alleles for the *FUL*, *FT*, and *PRR3* loci by comparing the flowering time data in Sanya, Nanning, and Beijing sites among landraces carrying multiple early-flowering allelic combinations. As expected, landraces carrying a relatively higher number of early-flowering alleles invariably exhibited

relatively earlier flowering times (Fig. 4f): a total of three landraces carried all the three early-flowering alleles, and these showed the earliest detected flower times, with the average maximum shortening effects for this set of three landraces being 51.25%, 54.73%, and 25.37% for the Sanya, Nanning, and Beijing sites, respectively (Fig. 4f). It should be noted that this apparently weaker shortening effect at the Beijing site was virtually certainly underestimated, as most of those landraces harboring no early-flowering alleles failed to bloom before the autumn frost. Collectively, these results highlight an opportunity to improve rice bean adaptability for growth in distinct latitudes through breeding efforts to combine the early-flowering alleles for three flowering time-controlling genes.

## The molecular basis and selection history of stem determinacy in cultivated rice bean

The stem determinacy trait is known to strongly influence lodging in legumes[65]. We collected data for stem determinacy traits in 2020 and 2021 for the 440 landraces at the Nanning site. The large majority (>85%) of the landraces exhibited an indeterminate stem growth phenotype (Supplementary Data 3). Notably, this distribution emphasizes that most rice bean landraces do not have the determinate stem growth phenotype that is amenable for mechanized cultivation systems. We performed GWAS analysis of stem determinacy based on the whole genome SNPs data for the germplasm panel and detected a total of 29 and 22 significant signals for stem determinacy in 2020 and 2021, including 7 signals detected repeatedly in both years (Supplementary Fig. 16; Supplementary Data 5; Supplementary Table 9). Among the repeatedly detected signals, the strongest signal was at

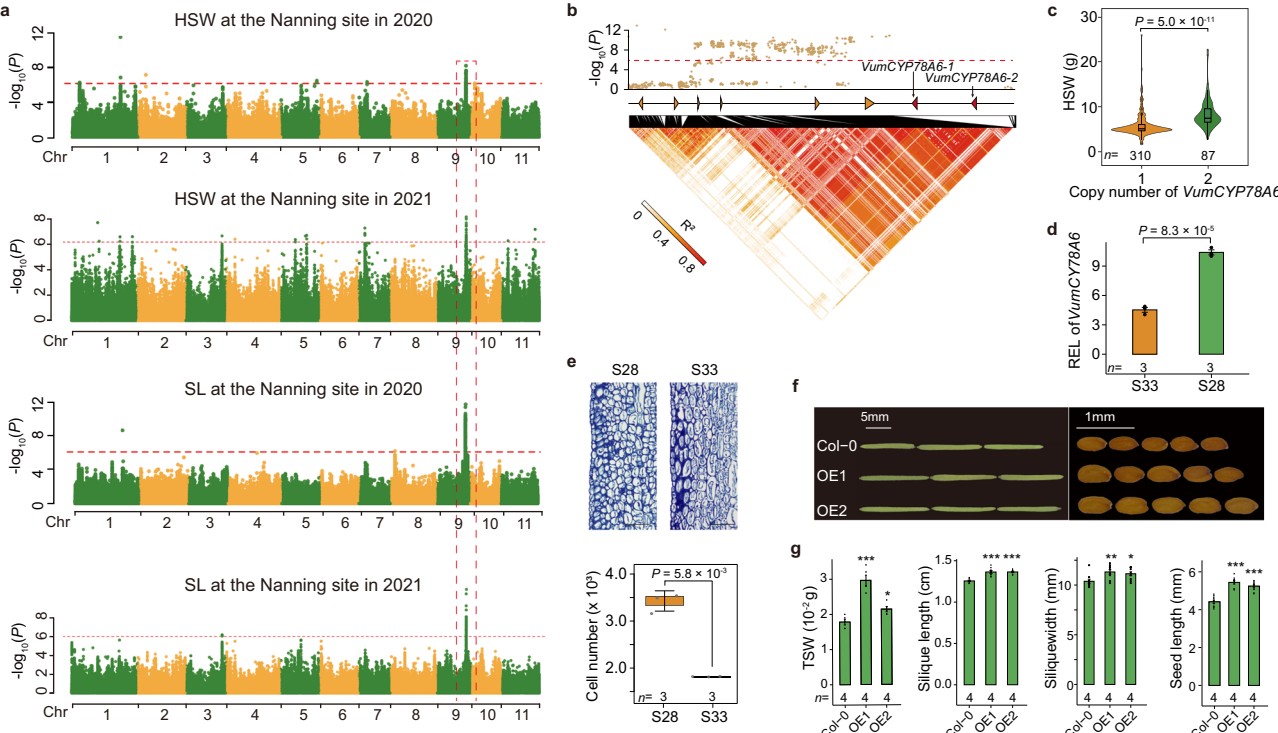

**Fig. 6 | Tandem duplication of the *VumCYP78A6* gene associated with seed yield traits. a** GWAS using the 2020 and 2021 Nanning datasets, indicating that the strongest association signals for hundred seed weight (HSW) and seed length (SL) traits all located at Chr9: 29,030,437–29,126,729. **b** Local Manhattan plot of HSW (top), the gene models (middle), and pairwise linkage disequilibrium heat map (bottom) at Chr9: 29,030,437–29,174,247. The two tandemly duplicated *Vum-CYP78A6* genes (*VumCYP78A6-1* and *VumCYP78A6-2*) are shown with the red dashed triangles. In **a** and **b**, the red horizontal dashed lines indicated the Bonferroni-corrected significance thresholds of GWAS ($\alpha = 1$). **c** The HSW distributions of landraces carrying distinct copy numbers of the *VumCYP78A6* gene. The number (*n*) of landraces carrying distinct copy numbers is shown below. **d** Bar plot showing the relative expression levels of *VumCYP78A6* in the pods at 16 DAP (days after pollination) from the long-seed landrace S28 (carrying two gene copies) and the short-seed landrace S33 (carrying one gene copy). **e** Light microscope images (top)

and cell number per square millimeter (bottom) of the cross-sections of the pod wall for the S28 and S33 landraces at 16 DAP. Scale bar, 100 µm. In the box plots of **c** and **e**, central line: median values; bounds of box: 25th and 75th percentiles; whiskers: 1.5*IQR (IQR: the interquartile range between the 25th and 75th percentile). **f** Silique (left) and seed (right) morphology of the wild type (Col-0) and two independent *Arabidopsis thaliana* transformants overexpressing the *VumCYP78A6-2* gene (OE1 and OE2). Scale bar: 5 mm for silique and 1 mm for seed. **g** The bar plots of thousand seed weight (TSW), silique length, silique width, and SL for Col-0, OE1, and OE2. *P* values are $1.07 \times 10^{-4}$, $1.11 \times 10^{-2}$, $2.36 \times 10^{-8}$, $4.18 \times 10^{-6}$, $1.44 \times 10^{-2}$, $5.25 \times 10^{-3}$, $1.88 \times 10^{-7}$, and $1.94 \times 10^{-6}$. The significance was tested using the two-sided Student's *t*-test in **c**, **d**, **e**, and **g**. *$P < 0.05$, **$P < 0.01$, ***$P < 0.001$ in (**g**). The data in **d** and **g** are shown as mean ± SE. In **d**, **e**, and **g**, the number (*n*) of each independent experiment is shown below. Source data are provided as a Source Data file.

Chr4 with high PVE values (17.96–41.43%) but spanned up to ~12 Mb genomic region (Chr4: 42,022,544–53,749,059).

The InDel-based GWAS for the 2-year stem determinacy data revealed a significantly associated InDel (2 bp–deletion at Chr4: 47,174,187) with a PVE of 22.01–35.21% positioned within the strongest SNP signal (Fig. 5a; Supplementary Data 5). Gene functional annotation revealed that this InDel apparently leads to premature termination of translation for the first exon in the gene *Vum_04G02513*, *TFL1* (*TERMINAL FLOWER1*; Fig. 5b), for which the ortholog in soybean was reported as the *Dt1* locus (*Gmtfl1* gene) controlling stem determinacy[25]. We found that a total of 32 landraces carried the homozygous mutation (2-nt deletion) alleles, which were identified using the sequencing data and were confirmed using Sanger sequencing (Supplementary Fig. 17). These landraces had a significantly higher proportion of determinate growth habit type compared to landraces harboring the reference alleles or the heterozygous alleles (Fig. 5c).

Notably, these 32 landraces were all in the SC group and were originally collected from an adjoining and mountainous area in South-Central China comprising five provinces (Chongqing, Hunan, Hubei, Guizhou, and Guangxi) (Fig. 5d). We also observed that these 32 landraces (represented by the bars with a predominant proportion of beige color in the Supplementary Fig. 18) were genetically distinct from other landraces within the SC group using model-based

clustering ($K = 4$), an inference that was further supported by a moderate level of differentiation ($F_{ST} = 0.11$). Notably, these landraces also displayed desirable agronomic traits including significantly earlier flowering time and significantly increased pod width, seed length, hundred seed weight, and branch number as compared to the other landraces of the SC group (Fig. 5e).

We next estimated the divergence time for these 32 landraces from the other landraces with distinct genetic admixture in the SC group inferred by the model-based clustering analysis (Fig. 3b), and obtained a similar divergence time of ~219 and 249 years ago using the SMC++[66] and MSMC2[67] methods, respectively (Fig. 5f; Supplementary Fig. 19). Our results collectively support that the 32 landraces carrying the homologous mutation alleles have been improved by producers in certain mountainous regions in South-Central China for at least 200 years, and suggest that these materials have huge potential for utilization in modern breeding programs seeking a variety of improvement goals.

## Tandem duplication of the *VumCYP78A6* gene associated with seed yield trait

Seed yield traits (including size and weight) have undergone strong selection in the domestication histories and modern breeding programs for legume crops[53,68–70]. We measured the hundred seed weight

(HSW) and seed length (SL) at the Nanning and Sanya sites in both 2020 and 2021. We next performed GWAS analysis to explore the genetic basis of these two traits, and identified one QTL locus significantly associated with the two traits at both examined sites in both examined years (Fig. 6a; Supplementary Fig. 20; Supplementary Table 9); this QTL is positioned at Chr9: 29,030,437–29,126,729, contains six predicted ORFs (Fig. 6b; Supplementary Data 7), and explains 5.99–16.17% of phenotypic variations (with the maximum value for SL at the Nanning site in 2020; Supplementary Data 5).

We next conducted a qPCR analysis for seed tissues of one long-seed landrace (S28) and one short-seed landrace (S33) at the 16 DAP (days after pollination) for the six candidate genes positioned within the aforementioned significantly associated interval on Chr9. Two of these genes showed significant differences in expression level between the two landraces, but neither of them had an obviously relevant functional annotation (Supplementary Fig. 21), which prompted us to explore the potential candidate genes positioned adjacent to this QTL. We found two tandemly repeated genes (*Vum_09G01129* and *Vum_09G01130*) at -10.45 kb downstream of the QTL (Fig. 6b). Using an in silico detection approach based on read depth information[71], a copy number variation (CNV) analysis of this gene in the 440 landraces showed that the 87 landraces carrying two copies exhibited significantly higher values for the two examined phenotypes than the 310 landraces carrying only one copy (Fig. 6c; Supplementary Fig. 22). These results suggested that the CNV may represent the causal variant controlling these two seed yield component traits.

The two duplicated genes had identical CDS sequences and were homologous to the *AtCYP78A6* gene (64.65% amino acid sequence identity; Supplementary Fig. 23), which encodes a cytochrome P450 monooxygenase known to function in maternally promoting seed growth by increasing the cell number in the integument of developing Arabidopsis seeds[72]. We, therefore, designated these rice bean genes as *VumCYP78A6-1* (*Vum_09G01129*) and *VumCYP78A6-2* (*Vum_09G01130*). A qPCR analysis showed that the expression level of *VumCYP78A6* in pod wall tissue at 16 DAP was significantly higher (-2-fold) in S28 than S33 (Fig. 6d). The impact of this CNV on the expression of *VumCYP78A6* was also verified in a larger panel comprising 20 landraces with one copy and 20 landraces with two copies. Specifically, qPCR analysis of the *VumCYP78A6* gene for the first fully expanded trifoliate leaves at 14 days after sowing showed a significantly ($P < 0.01$) higher expression level (-2-fold) in the 20 landraces with two copies than that in 20 landraces with one copy (Supplementary Fig. 24).

We also examined the number of cells in the pod wall at 16 DAP through cytological observation and detected a significantly increased number of cells in S28 compared to S33 (Fig. 6e). Finally, we generated two independent transgenic Arabidopsis lines by overexpressing the *VumCYP78A6-2* gene (Supplementary Fig. 25), both of which displayed significantly increased values for silique length, silique width, seed length, and seed weight (Fig. 6f, g; Supplementary Fig. 26). Viewed collectively, these results support *VumCYP78A6* as a highly probable causal gene underlying seed yield component traits in rice bean.

## Discussion

Rice bean has been proposed as a potential multipurpose legume crop to promote sustainable agriculture and fight hunger in Asia[18,73]. In the present study, we assembled a high-quality landrace FF25 reference genome and developed a valuable genomics resource by re-sequencing 440 rice bean landraces. By combining the high coverage of PacBio long reads and a Hi-C interaction map, our reference genome reached high accuracy and high continuity; this genome provides a valuable resource for future comparative genomics, evolutionary studies, and molecular research. Our rice bean genome assembly still contains 87 gaps, 78 of which have more than one flanking region (100 bp) with a high proportion (>90%) of repeat

sequences, suggesting that most of the gaps were caused by the incomplete assembly of the repeat sequences, which also reported by other studies[74,75]. We also predicted the candidate centromere regions using a previously published method[76] (see the "Methods" section) and found that all the 11 candidate centromere regions contained more than one assembly gap, suggesting none of the centromere sequences was fully assembled (Supplementary Table 11); future efforts using long sequencing reads (likely the ultra-long ONT reads) should help to 'close these gaps'. Additionally, our phylogenomic analysis clarified trends in the geographical distribution of the 440 rice bean landraces and revealed a bottleneck as well as an obvious "isolation by distance" pattern[77] for landraces during the northward dispersal of rice beans into and throughout China.

Genomic mutations associated with geographical adaptation allow the radiation of crop species to different agro-ecological and cultural environments[78]. Genetic control of flowering time is of great significance in determining the adaptation during the domestication and diversification of many crop species[79,80]. Appropriate timing to flowering is undoubtedly an advantage for survival and/or propagation[81] at distinct latitudes, as this impacts the growth period structure, yield, and quality of crops[82–84]. Studying flowering time is a large research field in plant biology because of its obvious agronomic implications, and studies from multiple species have shown that flowering time is controlled by multigene, highly topologically complex regulatory networks[85–87]. Our study has revealed how genetic alterations of the three known flowering loci—*FUL*, *FT*, and *PRR3*—have apparently supported rice bean's adaptation during its dispersal across a latitudinal gradient from South to North.

Agronomically, experience with crops including soybean and rice has established that flowering is delayed when a short-day crop species are grown at a high latitude location[84,88], so it is necessary to reduce the photoperiod sensitivity of such plants to advance flowering time and thus enable productive growth and yield[86]. We found that an early-flowering allele of the *PRR3* gene apparently supports early flowering for landraces from North of the Yellow river. It is notable that studies of barley[89], soybean[63,64], and rice[90] have also implicated *PRR* gene family members in high-latitude adaption. Previous studies in rice[91], cucumber[92], and soybean[93] have implicated natural variation in the *FT* gene in enhancing adaptation to higher latitudes. We identified an early-flowering allele of the *FT* gene that has apparently contributed to the adaptation of rice beans in the relatively low latitude region between the Yangtze and Yellow rivers. In contrast to these two alleles supporting adaptation to higher latitudes, a late-flowering allele of the *FUL* gene was found to have the potential to increase grain yield by extending the vegetative growth period and generating more branches at low latitude growth sites. Pleiotropy of the *FUL* gene has also been reported for other species including Arabidopsis[94], tomato[95], and *Setaria viridis*[96]. Similar to the *FUL* gene, in the short-day model plant soybean, breeding exploitation of the *J* gene (*ELF3*) has enabled the successful deployment of commercial soybean cultivation in tropical regions[97]. It is conceivable that—perhaps similar to successful efforts to variously combine mutations in four *E* loci in soybean[98]—our insights about the differential geographical distributions of alleles for flowering loci could be exploited to develop high-yield rice varieties for growth at low to high latitudes.

Our GWAS analyses helped decipher the genetic basis of stem determinacy in rice beans, detecting that stem determinacy of rice beans is influenced by the *TFL1* gene; this gene has been implicated in determining node termination and node number to control plant height and stem determinacy in many legumes species[25,99–101]. We also found that 32 landraces from Southern-Central China have multiple agronomically desirable traits and have undergone improvement by humans for at least 200 years; these materials should be considered for use as elite parents in rice bean breeding programs. Historically, elite landraces of other crops have been hugely beneficial to modern

breeding[26,102,103], for example with Taiwanese landraces in rice[104]: the so-called "miracle rice" IR8 with high yield supported the Green Revolution in Asia, and this line harbored a semi-dwarf allele from the Taiwanese landrace Dee-Geo-Woo-Gen[105]. Although QTLs for stem determinacy and seed yield-related traits were detected by our GWAS analyses in one and two environments respectively, further efforts should be made to investigate the robustness of these QTLs in more different environments.

Although the rice bean has been cultivated for thousands of years, to date it has received very little attention from breeders and agricultural scientists. The wealth of resources developed and identified in our study should help to rapidly advance breeding programs seeking to produce excellent varieties that simultaneously display geographically suitable flowering times, stem determinacy to support mechanized cultivation, and high yields through marker-assisted selection.

## Methods

### Plant materials and sequencing

The sequenced rice bean (*Vigna umbellata*) landraces used in this study were obtained from the Center for Crop Germplasm Resources, Institute of Crop Sciences, Chinese Academy of Agricultural Sciences, Beijing, China. An individual plant of rice bean landrace FF25 growing in a field in Beijing was used for the reference genome construction. The tender leaves were sampled for DNA extraction, and tissues including root, tender leaves, tender stem, flower, pod, and seed were harvested and immediately frozen in liquid nitrogen. Samples were stored at −80 °C prior to DNA or RNA extraction.

The high-quality genomic DNA from tender leaves was extracted and purified using DNeasy Plant Maxi Kits (Qiagen, Germany). The DNA concentration was measured using a NanoDrop spectrophotometer (Thermo Fisher Scientific, USA) and a Qubit 2.0 Fluorometer (Invitrogen, USA). Illumina short-read data were obtained using the Illumina NovaSeq platform, which generated a total of 338.19 million paired-end reads, with a total length of 50.73 Gb (Supplementary Table 1). Single-Molecule Real-Time (SMRT) cells were sequenced on the PacBio Sequel platform (Pacific Biosciences, USA), generating a total of 10.99 million reads with a total length of 142.95 Gb. Hi-C libraries were constructed from tender leaves using the Illumina NovaSeq platform. This allowed us to generate a total of 465.99 million paired-end reads and 69.90 Gb of sequencing data.

Each of the 440 landraces was planted at different sites for 2 years: (1) Beijing site (40.23°N, 116.56°E) with sowing date in the middle of June 2020 and 2021; (2) Nanning site (23.15°N, 108.28°E) with sowing date in the middle of July 2020 and 2021; (3) Sanya site (18.38°N, 109.21°E) with sowing date in the middle of November 2019 and 2020. Supplementary Fig. 27 presents the day length (per day) during the ~5-month growth period for the three sites. The day length differs obviously among the three sites (but was very similar between the two observation years). The average day lengths of Beijing during the first 4 months (during which all the landraces opened the first flower) was the longest (13.94 and 13.93 h) in both 2020 and 2021, followed by the Nanning site (12.52 and 12.53 h) and the Sanya site (11.28 and 11.28 h). Note that the phenotypes of the landraces grown at the Sanya site were measured the next year (i.e., 2020 and 2021; when the landraces were harvested); thus, the time of phenotypic data was designated as 2020 and 2021. For the plantings, 20 seeds of each landrace were sown in two rows (10 plants per row). Phenotypes in all three environments were investigated following the "Descriptors and data standards [*Vigna umbellate* (Thunb.) Ohwi & Ohashi]"[106]. Briefly, flowering time was recorded as the number of days after emergence (DAE) when the first flower opened. The main stem type was classified as indeterminate, semi-determinate, or determinate according to the growth state of the plants by observation[106] on five healthy individuals randomly selected from each plot for each landrace. The seed morphological

traits (seed length and hundred seed weight) for each landrace at each site were measured after harvest using automatic seed counting and analyzing instrument (Model SC-G, Hangzhou Wanshen Detection Technology Co., Ltd., Hangzhou, China, http://www.wseen.com/)[107]. The pod morphological traits (pod length and pod width) were measured using a vernier caliper with at least five healthy individuals for each landrace at each site after harvest.

### Genome assembly and quality assessment

In order to estimate the genome size of rice beans, the Illumina short reads were recruited to determine the *K*-mer distributions using GCE v1.0.2 (https://github.com/fanagislab/GCE). The PacBio long-read data were de novo assembled into PacBio contigs using Canu v1.9[34], and then the contigs were extended without the introduction of any gaps using the highly efficient repeat assembly (HERA) method[35], generating a total of 351 contigs with an N50 value of 18.26 Mb (Table 1). The Illumina short-read data was used for error-correcting of the contigs using Pilon[108]. Subsequently, to anchor the contigs into chromosomes, we aligned the Hi-C sequencing data into these contigs using Juicer v1.8.9[109]. The contigs were finally linked into 11 distinct chromosomes by 3D-DNA v180922[110].

The Illumina short-read data were also used to evaluate assembly accuracy and completeness using BWA-MEM v0.7.17-r118896[111]. The completeness of the genome assembly and the gene annotations were assessed with a plant database composed of 2121 conserved plant genes (eudicotyledons_odb10) using BUSCO v3.0.297[41].

### Repeats and gene annotation

The annotation of transposable elements was performed using RepeatMasker (http://www.repeatmasker.org). The repeat libraries included the RepBase-20170127 and a de novo repeat library created using RepeatModeler (http://www.repeatmasker.org) (with the parameter -LTRStruct). We analyzed the density distribution of the top-50 most abundant repeat subfamilies in 100 kb windows (using RepeatMasker), and used BEDtools[112] to merge the results with the parameter '-d 100000'. The rnd-6_family-604 subfamily (a 217-bp repeat) was identified as a centromere-specific repeat (Supplementary Fig. 28). The candidate centromere regions were also predicted according to the density distribution of this centromere-specific repeat (Supplementary Table 11). The LTRharvest[113] and the LTR_FINDER[114] programs were used to identify intact LTRs in the genomes of five *Vigna* species (*V. stipulacea*, *V. radiata*, *V. angularis*, *V. umbellata*, and *V. unguiculata*). LTR insertion times were estimated according to the formula $T = d/2m$ ($d$, the nucleotide distance for each pair of LTRs; $m$, the nucleotide substitution rate $= 1.64e^{-8}$).

Protein-coding genes were predicted using three different strategies: ab initio prediction, homology-based prediction, and transcript-based prediction. We used augustus[115] and SNAP[116] for ab initio predictions, and exonerate[117] was used for homology-based predictions. For transcript-based predictions, the RNA-Seq clean reads of tissues including root, tender leaves, tender stem, flower, pod, and seed were mapped to the genome assembly using HISAT2[118]. The mapping reads were assembled into transcripts using StringTie[119]. The transcripts were used for gene structure prediction using TransDecoder (http://transdecoder.github.io) and GeneMarkS-T[120]. These clean reads were also de novo assembled using Trinity[121] and the assembled transcripts were subsequently used for gene prediction using PASA[122]. Finally, EVidenceModeler (EVM) v1.1.1[123] was used to integrate the prediction results obtained by the above three methods (codon length ≥ 150 bp) to produce high-confidence gene models.

Ribosomal RNAs (rRNAs) were identified using RNAmmer[124] with default parameters. Reliable tRNA structures were detected using tRNAscan-SEM v1.23[125]. Non-coding RNAs containing miRNA and snoRNA features were annotated using INFERNAL[126] with default parameters. Pseudogenes were identified using the published

pipeline[127]. The transcription factors and transcription regulators were annotated using iTAK v18.12[128] with default parameters.

## Gene families and phylogenetic analysis

We used OrthoFinder v2.3.9[129] to identify shared gene families between rice beans and 13 other plant species, including five *Vigna* species (*V. stipulacea*, *V. radiata*, *V. angularis*, *V. umbellata*, and *V. unguiculata*), four other legumes (*Phaseolus vulgaris*, *Glycine max*, *Lotus japonicus*, and *Arachis duranensis*), five other eudicots (*Arabidopsis thaliana*, *Citrus sinensis*, *Populus trichocarpa*, *Vitis vinifera*, and *Solanum lycopersicum*), and one monocot (*Oryza sativa*). Based on the protein sequences of 334 single-copy ortholog families, the phylogenetic relationships among these species were estimated using RAxML v8.2.12[130]. Divergence times were estimated by the MCMCtree program embedded in PAML v4.9[131]. We measured the expansion and contraction of orthologous gene families based on a maximum likelihood tree using CAFE v4.2 (https://github.com/hahnlab/CAFE).

## KEGG enrichment analysis

The R package ClusterProfiler v3.18.0[132] was used to perform KEGG enrichment analysis. KEGG terms showing adjusted *P* values < 0.05 were considered significantly enriched.

## Comparative genomics and Ks analysis

Gene synteny analysis was performed using MCScanX[43] and BLASTP[133] (−evalue < 1e−10, -v 5, -b 5) to determine the pairwise similarity among the protein sequences of *Glycine max*, *Phaseolus vulgaris*, and five *Vigna* species (*V. stipulacea*, *V. radiata*, *V. angularis*, *V. umbellata*, and *V. unguiculata*). The synteny figure was plotted using the NGenomeSyn program (https://github.com/hewm2008/NGenomeSyn). Synonymous nucleotide substitutions on synonymous sites (Ks) were estimated using the WGDi tool (https://github.com/SunPengChuan/wgdi) with default parameters.

## SNP and small InDel calling

We sequenced the genomes for 440 rice bean landraces with an average depth of 24.91× using the Illumina NovaSeq platform (Supplementary Data 3). The quality control for the raw sequencing data was performed using fastp v0.20.1[134] with default settings. The high-quality short reads were aligned to the genome using BWA-MEM v0.7.17-r118896[111]; PCR duplicates were removed using Picard v1.118 (http://broadinstitute.github.io/picard/); SNPs and InDels were identified using HaplotypeCaller of the Genome Analysis Toolkit (GATK) v4.1.5.0[135], and were subsequently filtered ('QD < 2.0 || FS > 60.0 || MQ < 40.0 || MQRankSum < −12.5 || ReadPosRankSum < −8.0' for SNPs, and 'QD < 2.0 || FS > 200.0 || ReadPosRankSum < −20.0' for InDels)[49]. Non-biallelic SNPs/InDels with a read depth < 5 were removed from further analyses.

## Phylogenetic and population structure analyses

A total of 1,400,862 SNPs with a minor allele frequency (MAF) ≥ 0.05 and missing rate ≤ 50% were used to build a maximum likelihood phylogenetic tree using TreeBeST v1.9.2[136], as well as to perform principal component analyses (PCA) using the smartPCA program embedded in the Eigensoft package v7.2.1[137]. The $\pi$ and $F_{ST}$ values were calculated using VCFtools v0.1.17[138] based on the same SNP set. Population structure was investigated based on 20,000 randomly selected SNPs using STRUCTURE v2.3.4[50] with 100,000 iterations of burning and 200,000 iterations of MCMC, and evaluating each *K* from 2 to 4.

## Divergence time estimation

MSMC2 v2.1.1[67] was used to infer the divergence times of stem determinacy landraces carrying homologous deletion mutation alleles with other landraces carrying the homologous reference alleles in the SC group. To improve reliability, genome regions were masked with the SNPable tool (http://lh3lh3.users.sourceforge.net/snpable.shtml) when the coverage depth was <15× after removing reads with mapping quality <20. First, we split the reference genome into overlapping 35-mers and then mapped these 35-mers back to the reference genome using BWA[139] (bwa aln -R 1000000 -O 3 -E 3). Only regions where the majority of 35-mers were uniquely mapped and without mismatch were retained for further analysis. We selected the top 10 samples in each population with the highest coverage after masking. The 8 most frequent haplotypes were randomly selected from the 10 samples in order to infer the demographic history of each population. We repeated this procedure 20 times. Scaled times were converted to years by assuming a generation time of 1 year and a mutation rate of $\mu = 1.5 \times 10^{-8}$ per site per generation[140]. We also used the SMC++ v1.15.2[66], which does not rely on haplotype phase information, to estimate the divergence times (using the same generation time and mutation rate).

## Linkage disequilibrium

To estimate and compare the patterns of linkage disequilibrium (LD) decay in each population, we computed the mean squared correlation coefficient ($r^2$) values between any two SNPs within 300 kb using PopLDdecay v3.41[141].

## GWAS analysis

We retained SNPs with a MAF ≥ 0.05 and a missing rate ≤ 50% to perform GWAS analysis. After imputation using Beagle v4.1[142] with default parameters, the GWAS analysis was performed based on a linear mixed model using the program Fast-LMM v2.06.20130802[143]. The *P* value threshold for significance was estimated as $1/n$ (where *n* corresponds to the SNP number). The phenotypic variance that was explained by each SNP was estimated following the below previously reported method[144]:

$$\text{PVE} = \frac{2\hat{\beta}^2 \times \text{MAF} \times (1 - \text{MAF})}{2\hat{\beta}^2 \times \text{MAF} \times (1 - \text{MAF}) + \left(\text{se}\left(\hat{\beta}\right)\right)^2 \times 2N \times \text{MAF} \times (1 - \text{MAF})}$$

$$(1)$$

where $\hat{\beta}$ and MAF is the effect size estimate and minor allele frequency for the SNP, $N$ is the sample size, and $\text{se}(\hat{\beta})$ is the standard error of effect size for the SNP.

## Inference of the inheritance model for alleles of the flowering time genes

To infer the most likely inheritance model of alleles for the flowering-time related loci (*FUL*, *FT*, and *PRR3*), we used the R package "SNPassoc"[145] to perform association analysis of the alleles based on several genetic models (co-dominant, dominant, recessive, over-dominant, or additive). The model with the smallest Akaike information criteria (AIC) value was identified as the best fitting genetic model.

## Histological analysis

Cross-sections of pod walls from S28 and S33 landraces were analyzed by light microscopy (BX51; Olympus). The pod wall tissues were sampled at 16 DAP and immediately fixed with FAA: glacial acetic, 38% form aldehyde, 70% ethanol (1:1:18), and then dehydrated through a standard ethanol series. The pod wall tissues were embedded in Paraplast Plus tissue-embedding medium (Sigma-Aldrich), sectioned at 8 mm using a microtome (RM2235, Leica Microsystems), and then stained with toluidine blue. The cell numbers in the cross-sections were measured using Olympus Stream software. The analysis was based on at least three biological replicates.

## RNA extraction and qPCR analysis

qPCR analysis was used to quantify the relative expression levels of the *FUL* gene in newly expanded leaves in a panel of 16 diverse landraces, the expression levels of the *VumCYP78A6* gene in the seed and pod tissues of the S28 and S33 landraces, and the expression levels of the *VumCYP78A6-2* gene in the primary inflorescence stems of the transgenic Arabidopsis plants. Total RNA was extracted using Trelief™ RNAprep Pure Plant Kits (Polysaccharides & Polyphenolics-rich) (Tsingke, China). First-strand cDNA was synthesized using a PrimeScript™ RT Reagent Kit with gDNA Eraser (Takara, Japan). Quantitative PCR was performed using TSINGKE Master qPCR Mix (SYBR GreenIwith UDG) (Tsingke, China), on a StepOnePlus™ Real-Time PCR System (Applied Biosystems, USA) following the manufacturer's instructions. cDNA transcript levels were normalized to those of the reference gene *ACTIN* using the $2^{-\Delta\Delta CT}$ method[146]. PCR reactions were performed in triplicate for each biological replicate; three or more biological replicates were assessed. Primers were designed to span an intron in order to avoid the amplification of genomic DNA and are shown in Supplementary Table 12.

## Arabidopsis transformation

The total RNA of the pod tissue from the FF25 landrace was extracted and reverse transcription was performed. The full coding sequence of the *VumCYP78A6-2* gene (*Vum_09G01130*) was amplified and cloned into the pEasy-T1 vector. The binary vector pCambia3301 was used to subclone the gene for overexpression. The construct was individually introduced into *Agrobacterium tumefaciens* strain GV3101 and transformed into the Arabidopsis ecotype Columbia (Col-0) using the floral dip method[147]. Relative expression levels of the *VumCYP78A6-2* gene in primary inflorescence stems of 2-week-old T1 transgenic plants were measured with qPCR, and two lines with relatively high *VumCYP78A6-2* expression were selected for further analyses. All phenotypes were measured for T3 homozygote plants. Primers are shown in Supplementary Table 12.

## Reporting summary

Further information on research design is available in the Nature Research Reporting Summary linked to this article.

## Data availability

Data supporting the findings of this work are available within the paper and its Supplementary Information files. A reporting summary for this Article is available as a Supplementary Information file. The datasets and plant materials generated and analyzed during the current study are available from the corresponding author upon request. All datasets reported in this study have been deposited in the National Center for Biotechnology Information (NCBI) with the following accession IDs: FF25 genome assembly, JALEER000000000; Raw data for FF25 genome assembly, PRJNA819955; Raw data for genome sequencing of 440 landraces, PRJNA803965. The annotation files including predicted CDS and protein sequences generated for FF25 genome assembly have been deposited at Figshare [https://doi.org/10.6084/m9.figshare.19420058]. The online tools and database used in this paper include: Pfam [http://pfam.xfam.org/], InterPro [https://www.ebi.ac.uk/interpro], NR [https://www.ncbi.nlm.nih.gov/refseq/about/nonredundantproteins/], GO [http://geneontology.org], KEGG [https://www.genome.jp/kegg/], FLOweRing Interactive Database [http://www.phytosystems.ulg.ac.be/florid/]. Source data are provided with this paper.

## Code availability

The scripts used for the analyses can be freely and openly accessed at GitHub [https://github.com/guanjiantao-caas/Code-for-rice-bean].

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

## Acknowledgements

This work was supported by the National Key Research and Development Program of China (grant nos. 2019YFD1001303 and 2019YFD1001300), the China Agriculture Research System of MOF and MARA (CARS-08), the Agricultural Science and Technology Innovation Program (ASTIP) from Chinese Academy of Agricultural Sciences, the Program of Protection of Crop Germplasm Resources in China (grant nos. 2019NWB036-07 and 19200385-6), the Third National General Survey and Collection of Crop Germplasm Resources (grant no. 19200354), and the Program Management Unit for Human Resources and Institutional Development, Research and Innovation of Thailand (grant no. B16F640185).

## Author contributions

L.X.W. and H.X. designed the research. L.X.W., P.S., K.L., X.Z.C., S.H.W., H.L.C., and Y.L.W. provided materials and information. J.T.Z., G.L.L., S.H.W., Y.H.C., Y.W. Z., X.X.Y., X.C., and A.H.S. contributed to phenotyping. J.T.G. and Z.Q.Z. conducted genome assembly, gene annotation, and population analyses. D.G., Z.H., and Y.Y. performed the experiments. J.T.G. and L.X.W. wrote and revised the manuscript with input and comments from the other authors. All authors read and approved the manuscript.

## Competing interests

The authors declare no competing interests.

 

## Additional information

[1]Institute of Crop Sciences, Chinese Academy of Agricultural Sciences, Beijing, China. [2]Institute of Biotechnology, Beijing Academy of Agriculture and Forestry Sciences, Beijing, China. [3]Institute of Vegetables and Flowers, Chinese Academy of Agricultural Sciences, Beijing, China. [4]College of Agriculture, Yangtze University, Jingzhou, China. [5]Institute of Rice Research, Guangxi Academy of Agricultural Sciences, Nanning, China. [6]Department of Agronomy, Faculty of Agriculture at Kamphaeng Saen, Kasetsart University, Nakhon Pathom, Thailand. [7]Institute of Industrial Crops, Jiangsu Academy of Agricultural Sciences, Nanjing, China. [8]College of Agriculture, Shanxi Agricultural University, Taiyuan, China. [9]Crop Research Institute of Hunan Province, Changsha, China. [10]These authors contributed equally: Jiantao Guan, Jintao Zhang, Dan Gong. ✉e-mail: xiehua@baafs.net.cn; wanglixia03@caas.cn

