## [Peer Review File · Nature Communications]

Genomic analyses of rice bean landraces reveal adaptation and yield related loci to accelerate breedingReviewers' Comments:

Reviewer #1:

Remarks to the Author:

This paper de novo assembled the first high quality reference genome in rice bean and its genomic features and evolutionary comparisons with other legumes were analyzed. Using this novel reference genome, 440 diverse landraces were re-sequenced. Some important agronomic traits were phenotyped in three different environments and some candidate genes were identified through GWAS combining LD associations. In particular, three flowering genes, FUL, FT and PRR3 were proposed as the candidates for the local adaptations to different latitude in rice bean. As other legume crops, loss of function of orthologue of TFL1 contributed to the determinate growth habit and shortened plant height in rice bean. Most interestingly, copy number variation of a cytochrome P450 monooxygenase was shown to control yield traits like hundred seed weight, pod length and pod width. Overall, this paper offered a lot of valuable resources for high quality reference genome, re-sequenced genomic data for the core germplasm collections, also some candidate genes for flowering and adaptation, growth habit and yield traits. However, I have some major concerns which should be addressed to further improve this manuscript.

1. GWAS is highly dependent on the in the population structure, population size, and consistency of phenotypes in which phenotypes are largely influenced by environments. Unfortunately, in this paper, all the phenotypes are collected from three independent environments, Sanya in 2021, Nanning in 2020, and Beijing in 2021 without replications of these three environments in different years. All the GWAS results are from single environment without replication which makes the results unreliable, risky and very doubtful. Therefore, I strongly suggested that all the phenotypes must be repeated in these three locations (Sanya, Nanning, Beijing) in 2022 and perform GWAS again to validate the QTNs detected before. If each of the QTNs in FUL, FT, PRR3, TFL1 and VumCYP78A6-1 can be repeatedly and consistently detected in each location/environment, the further functional analysis should have meanings and significances, otherwise these QTNs are the false detections.
2. The 440 accessions collected from South Asia and different provinces in China, please indicated the exact country or province where these landraces were originated or collected. These information is critical to understand the latitude adaptation and photoperiod flowering.
3. There are no any descriptions in the method about how the phenotypes had been investigated in three environments/locations. What the time had the landraces been sowed? Different sowing time plant will have different phenotypes since photoperiod is different in between Spring sowing and Autumn sowing. How many plants were evaluated from each landrace?
4. In Fig4, early flowering allele and late-flowering allele were presented for all three genes FUL, FT, and PRR3 without detailed sequence polymorphism or sequence variations which reduced the creditability and confidence of the candidate genes. Please specify the sequence variations (promoter variations caused transcriptional changes or CDS variations caused amino acid changes) for each early or late allele for all the candidate genes, FUL, FT, and PRR3.
5. Within three locations, Sanya and Nanning are the short-day environments. But the GWAS results for flowering time are not consistent suggested the signals of FT and FUL are very doubtful as I mentioned in my first concern. If FT can be detected in Nanning, it should be detected in Sanya. Similar, FUL detected in Sanya and it should be detected in Nanning as well. These GWAS results must be detected in multiple locations or multiple years to validate their authenticity.
6. In the field conditions, the determinate, semi-determinate or indeterminate may not easily to be discriminated due to the genetic epistasis or environmental effects. In legumes (pea, soybean, mung bean), I believe it is same in rice bean, TFL1 determines the node termination and node numbers to control plant height. Plant height can be separated by node number (controlled by TFL1) and internode length (normally controlled by GA). I wonder whether the node numbers of 440 landraces had been investigated and GWAS should be analyzed which is more precisely to represent the effect of TFL1.
7. In relation of question 6, I suggest to divide population in Fig5 e, f into 2 nt deletion and non deletion of TFL1, other than Det and Indet. Because the phenotyping of Det and Indet may not

accurate.

8. It is very interesting to identify that the copy number of CYP78A6 is related to yield trait of rice bean. CNV always causes the dosage effect of the gene duplicated which can be reflected by transcriptional levels. Therefore, the expression levels of random selected landraces, for example to check the expressions of 20 landraces with one copy and 20 landraces with two copies, and compare the transcriptional differences between two groups.

Reviewer #2:

Remarks to the Author:

The authors present the reference-quality genome assembly of *Vigna umbellata*, an underutilized legume crop from Asia. I agree with the authors that this represents a relevant resource that will facilitate accelerated and more targeted breeding of rice bean, with some potential in local diet. The genome assembly is of good quality and contiguity, with the relevant parameters (BUSCO, LAI, N50 etc) in an appropriate range. There seems to be a portion of sequence missing from the assembly, as compared to the estimated genome size (and also visible in the BUSCO scores). It would be helpful for the reader to learn whether this sequence is potentially located/missing in the centromeric regions. How well are the centromeres represented/resembled in this assembly?

The collinear genes/segments displayed in the Circos illustration (Figure 1b; track VIII) suggest that there is quite a large number of those...how does this go together with the smaller % of duplicated genes in the BUSCO evaluation and the conclusion that no recent WGD was detected? What were the criteria for collinearity in Figure 1b? Further, in line 148, how many "syntenic blocks" and homologous gene pairs were identified, and how?

Figure 3b: what is the beige/brown color? Background?

I do have some concerns/questions with the observed flowering time variation that the authors relate to latitude. These data are from different years, without any replicates or controls as far as I understand. How sure can you be that the observed differences are not simply due to climate/temperature differences or other environmental factors in the years/locations? I understand/assume all 440 landraces been grown in all three locations?

The GWAS identified candidate genes appear plausible to me, and the conclusions for latitude and allele analyses is interesting, but I would like to see some kind of control for the flowering time variation.

Another major concern is with data availability: I could not access any (!) of the given accession codes or figshare download links. I understand authors may want to wait until publication but should provide a possibility for data download and inspection during reviewing. As a result I cannot comment on data quality, completeness and presentation.

Reviewer #3:

Remarks to the Author:

The authors report the high-quality genome of *Vigna umbellata* and the analysis of germplasm population. The approach and methods analysis are gold standard and resulted in outstanding results. The authors revealed important genes associated with agronomic traits from GWAS analysis of the germplasm. There are only minor points to be added to make the manuscript even stronger.

1. Gene synteny block analysis between *Vigna* species would demonstrate how genome structure and order of gene are conserved in this genus.

2. Selection (K_a/K_s) between tropical and temperate legumes would be interesting: *V. umbellata* is widespread across both tropical and temperate regions.

3. Flowering time: Please describe photoperiod of the three locations (Sanya, Nanning, Beijing) during the time of experiment performed, describe action mode whether these alleles (FUL, FT, PRR3) are dominant, recessive or co-dominant.

4. Fig2A: misspelling of *Vigna*

Reviewer #1 (Remarks to the Author):

This paper de novo assembled the first high quality reference genome in rice bean and its genomic features and evolutionary comparisons with other legumes were analyzed. Using this novel reference genome, 440 diverse landraces were re-sequenced. Some important agronomic traits were phenotyped in three different environments and some candidate genes were identified through GWAS combining LD associations. In particular, three flowering genes, *FUL*, *FT* and *PRR3* were proposed as the candidates for the local adaptations to different latitude in rice bean. As other legume crops, loss of function of orthologue of *TFL1* contributed to the determinate growth habit and shortened plant height in rice bean. Most interestingly, copy number variation of a cytochrome P450 monooxygenase was shown to control yield traits like hundred seed weight, pod length and pod width. Overall, this paper offered a lot of valuable resources for high quality reference genome, re-sequenced genomic data for the core germplasm collections, also some candidate genes for flowering and adaptation, growth habit and yield traits. However, I have some major concerns which should be addressed to further improve this manuscript.

1. GWAS is highly dependent on the in the population structure, population size, and consistency of phenotypes in which phenotypes are largely influenced by environments. Unfortunately, in this paper, all the phenotypes are collected from three independent environments, Sanya in 2021, Nanning in 2020, and Beijing in 2021 without replications of these three environments in different years. All the GWAS results are from single environment without replication which makes the results unreliable, risky and very doubtful. Therefore, I strongly suggested that all the phenotypes must be repeated in these three locations (Sanya, Nanning, Beijing) in 2022 and perform GWAS again to validate the QTNs detected before. If each of the QTNs in *FUL*, *FT*, *PRR3*, *TFL1* and *VumCYP78A6-1* can be repeatedly and consistently detected in each location/environment, the further functional analysis should have meanings and significances, otherwise these QTNs are the false detections.

Response:

We would first like to thank the Reviewer for the supportive review and for the very helpful guidance about how to improve our study.

Regarding this comment specifically, to be clear, we need to stress that we do have multi-year evidence for all of the associations that we present in our study. Specifically, we have two-year (2020 and 2021) phenotyping data for traits including flowering time at all three sites, for stem determinacy at the Nanning site, and for both HGW and seed length at the Sanya and Nanning sites. GWAS of this phenotype data revealed 15 QTLs that were repeatedly detected across two years (containing candidate genes including *FUL*, *FT*, *PRR3*, *TFL1*, and *VumCYP7&A6*) (Supplementary Table 9). Note that we have added Supplementary Figures and a Table to present the Manhattan plots of GWAS for flowering-time data (Supplementary Fig. 8), seed yield traits (Supplementary Fig. 20), stem determinacy (Supplementary Fig. 16), as well as the detailed information for all associated loci (Supplementary Data 5).

Kindly note that in the originally submitted manuscript, we did have claims of associations for three traits that lacked multi-year evidence (plant height at the Nanning site in 2020; pod length and pod width at the Nanning site in 2020 and the Sanya site in 2021). These have been dropped from the revised manuscript.

Supplementary Table 9. GWAS QTLs detected across two years (2020 and 2021)

Trait	Site	QTL	Years ^a	PVE (%)	Candidate gene
Flowering time	Sanya	Chr8:1,909,300-1,919,793	2	10.57	-
		Chr11:6,142,933-6,162,249	2	14.86	Vum_11G00418 (FUL)
	Nanning	Chr4:35,931,101-35,996,258	2	8.23	Vum_04G01668 (FT)
	Beijing	Chr2:38,647,190-38,647,190	2	17.30	Vum_02G01965 (PRR3)
Stem determinacy	Nanning	Chr3:12,535,769-16,852,788	2	8.69	-
		Chr4:42,022,544-54,020,700	2	41.43	Vum_04G02513 (TFL1)
		Chr5:13,477,451-13,789,216	2	32.19	-
		Chr6:2,759,250-6,161,777	2	12.42	-
		Chr7:18,729,386-20,781,885	2	8.02	-
		Chr9:28,394,569-28,697,545	2	8.24	-
		Chr11:5,078,886-15,461,758	2	24.72	-
HGW	Sanya	Chr9:29,081,975-29,126,729	2	8.88	Vum_09G01129 &
	Nanning	Chr9:29,030,437-29,126,729	2	15.74	Vum_09G01130
Seed length	Sanya	Chr9:29,030,437-29,126,729	2	11.39	(VumCYP7&A6-1 &
	Nanning	Chr9:29,030,437-29,126,729	2	16.17	VumCYP7&A6-2)

^aNumber of years in which the QTL was detected.

Supplementary Fig. 8. Manhattan plots of GWAS for flowering-time data measured at the Sanya (a), Nanning (b), and Beijing (c) sites across two years (2020 and 2021).

Supplementary Fig. 20. Manhattan plots of GWAS for seed yield traits measured at the Nanning (a and b) and Sanya (c and d) sites across two years (2020 and 2021). HSW, hundred seed weight; SL, seed length.

Supplementary Fig. 16. Manhattan plots of SNP-based (a) and InDel-based (b) GWAS for stem determinacy measured at the Nanning site across two years (2020 and 2021). The peak InDel (2-nt deletion) is indicated by the red circle.

2. The 440 accessions collected from South Asia and different provinces in China, please indicated the exact country or province where these landraces were originated or collected. These information is critical to understand the latitude adaptation and photoperiod flowering.

Response: Thanks for your comment. We have now provided the province and/or country information for where the landraces were collected in Supplementary Data 3 in our revised manuscript.

3. There are no any descriptions in the method about how the phenotypes had been investigated in three environments/locations. What the time had the landraces been sowed? Different sowing time plant will have different phenotypes since photoperiod is different in between Spring sowing and Autumn sowing. How many plants were evaluated from each landrace?

Response: Thanks for your comment. We have now added this information to the ‘Methods’ section of our revised manuscript. The revised content reads as follows (Lines 477-499):

“Each of the 440 landraces were planted at different sites for two years: (1) Beijing site (40.23° N, 116.56° E) with sowing date in the middle of June 2020 and 2021; (2) Nanning site (23.15° N, 108.28° E) with sowing date in the middle of July 2020 and 2021; (3) Sanya site (18.38° N, 109.21° E) with sowing date in the middle of November 2019 and 2020. Supplementary Fig. 27 presents the day length (per day) during the ~ five month growth period for the three sites. The day length differs

obviously among the three sites (but was very similar between the two observation years). The average day lengths of Beijing during the first four months (during which all the landraces opened the first flower) was the longest (13.94 and 13.93 h) in both 2020 and 2021, followed by the Nanning site (12.52 and 12.53 h) and the Sanya site (11.28 and 11.28 h). Note that the phenotypes of the landraces grown at the Sanya site were measured the next year (*i.e.*, 2020 and 2021; when the landraces were harvested); thus, the time of phenotypic data was designated as 2020 and 2021. For the plantings, 20 seeds of each landrace were sown in two rows (10 plants per row). Phenotypes in all the three environments were investigated following the previously published “Descriptors and data standards [*Vigna umbellata* (Thunb.) Ohwi & Ohashi]”¹⁰⁷. Briefly, flowering time was recorded as the number of days after emergence (DAE) when the first flower opened. The main stem type was classified as indeterminate, semi-determinate, or determinate according to the growth state of the plants by observation¹⁰⁷ on five healthy individuals randomly selected from each plot for each landrace. The seed morphological traits (seed length and hundred seed weight) for each landrace at each site were measured after harvest using a previously described¹⁰⁸ automatic seed counting and analyzing instrument (Model SC-G, Hangzhou Wanshen Detection Technology Co., Ltd., Hangzhou, China, <http://www.wseen.com/>). The pod morphological traits (pod length and pod width) were measured using a vernier caliper with at least five healthy individuals for each landrace at each site after harvest.”

Supplementary Fig. 27. The day length (hour) per day during the growth period at Sanya site (18° N) in 2019-2020 and 2020-2021, at Nanning site (22° N) in 2020 and 2021, and at Beijing site (40° N) in 2020 and 2021.

4. In Fig4, early flowering allele and late-flowering allele were presented for all three genes *FUL*, *FT*, and *PRR3* without detailed sequence polymorphism or sequence variations which reduced the creditability and confidence of the candidate genes. Please specify the sequence variations (promoter variations caused transcriptional changes or CDS variations caused amino acid changes) for each early or late allele for all the candidate genes, *FUL*, *FT*, and *PRR3*.

Response: Thanks for your comment. We have now added the detailed sequence polymorphism information for these genes to the revised manuscript (Supplementary Fig. 10, 13 and 14). We extracted the significantly associated SNPs and InDels from the upstream (< 30 Kb) to downstream (< 30 Kb) regions of the *FUL*, *FT*, and *PRR3* candidate genes.

For *FUL*, we have now added new content in the revised manuscript as follows (Lines 244-249):

“All the significantly associated SNPs and InDels in this GWAS signal were located in its upstream region (> 5 Kb) (Supplementary Fig. 10), suggesting that these polymorphisms could influence *FUL* expression to control flowering time. This was further supported by the observation that the expression level of *FUL* (in newly expanded leaves in a panel of 16 diverse rice bean landraces) was strongly negatively correlated ($R = -0.69$, $P < 0.01$) with flowering time (Supplementary Fig. 11).”

For *FT*, we have now added new content in the revised manuscript as follows (Lines 255-258):

“We found two significantly associated SNPs around (< 2 Kb) and within *FT* gene; one SNP (Chr4:35,950,445) was located upstream (< 200 bp) of the transcription start site and another was located in the first intron (Chr4:35,951,311) (Supplementary Fig. 13).”

For *PRR3*, we found nine associated polymorphisms positioned in the upstream region (< 2 kb; four SNPs and two InDels), first intron (one SNPs), third intron (one SNP) and third CDS (one SNP). The

SNP in the third CDS was the peak SNP (C/T), which encodes a nonsynonymous ('S' to 'F') mutation in the functional PR (pseudo receiver) domain. Thus, we have now added new content in the revised manuscript as follows (Lines 259-267):

“For the Beijing data, we repeatedly detected a peak SNP in the *PSEUDO-RESPONSE REGULATOR 3 (PRR3)* gene (*Vum_02G01965*) at Chr2: 38,647,190 (7.30% – 17.30% of PVE), encoding a nonsynonymous variant ('S' to 'F') in the third CDS consisting of the functional PR (pseudo receiver) domain (Fig. 4c; Supplementary Data 5; Supplementary Fig. 8; Supplementary Fig. 14; Supplementary Table 9). *PRR3* is an ortholog of the known soybean circadian clock gene *GmTof12/GmPRR3b* that has been previously shown to function as a major flowering time regulatory gene and has been linked to the expansion of soybean into higher latitudes^{64,65}. Notably, a similar effect from a single amino acid change ('S' to 'L') on flowering time has also been reported for the *GmPRR3b* gene in soybean⁶⁴.”

Supplementary Fig. 10. Local Manhattan plot for SNPs and InDels significantly associated with flowering time at the Sanya site within 30 Kb of the candidate gene *FUL*.

Supplementary Fig. 13. Local Manhattan plot for SNPs and InDels significantly associated with flowering time at the Nanning site within 30 Kb of the candidate gene *FT*.

Supplementary Fig. 14. Local Manhattan plot for SNPs and InDels significantly associated with flowering time at the Beijing site within 30 Kb of the candidate gene *PRR3*. The linear gene structure displays the CDS region (rectangles), and introns (horizontal solid grey lines). The white arrow indicates the 5' to 3' direction. The red dot and vertical solid lines represent the peak SNP. The dotted lines represented the physical coordinates of the two functional domains (PR, pseudo receiver; CCT, named after the proteins CONSTANS [CO], CO-like, and TOC1, which contain this domain).

To avoid confusion, we have added the allele information for the early-flowering and late-flowering alleles of *FUL*, *FT*, and *PRR3* loci in Fig.4 of the revised manuscript as follows:

Fig. 4 | The genetic architecture underlying flowering time control from low to high latitudes. a-c, Manhattan plots of GWAS for flowering-time data measured in Sanya (18° N) in 2021 (a), Nanning (22° N) in 2020 (b), and Beijing (40° N) in 2021 (c). Pie charts represented allelic frequencies of the major associated loci. The bar plots display the flowering time of landraces carrying each allele of the identified major loci. DAE, days after emergence. The significance of flowering time between landraces carrying distinct alleles was tested with Wilcoxon tests. **d,**

The geographical distributions of landraces carrying the early-flowering alleles of *FT* and *PRR3*, and late-flowering allele of *FUL*, respectively. The violin plot showed the significant differences of latitudes between landraces carrying the early-flowering allele of *FT* and *PRR3*, respectively. The significance was tested with Wilcoxon tests. In the box plots, central line: median values; bounds of box: 25th and 75th percentiles; whiskers: 1.5 * IQR (IQR: the interquartile range between the 25th and 75th percentile). e, The bar plots show the flowering shortening effects of the early-flowering alleles of *FT* and *PRR3* at each of the three measurement sites. NA indicates landraces carrying neither of these two early-flowering alleles. f, The dot plots show the flowering time shortening effects of early-flowering allelic combinations at each of the three measurement sites. Blue dots represent the landraces categorized according to all of the different allelic combinations found in the 440 sequenced landraces. Red lines indicate the average value of each category. NA indicates landraces carrying no early-flowering alleles. The number of landraces for each category is shown below.

5. Within three locations, Sanya and Nanning are the short-day environments. But the GWAS results for flowering time are not consistent suggested the signals of *FT* and *FUL* are very doubtful as I mentioned in my first concern. If *FT* can be detected in Nanning, it should be detected in Sanya. Similar, *FUL* detected in Sanya and it should be detected in Nanning as well. These GWAS results must be detected in multiple locations or multiple years to validate their authenticity.

Response: Thanks for your comment. As we responded above, all of the associations reported in the revised manuscript have data and GWAS analysis for multiple years. For the Sanya and Nanning sites, two and one flowering-time related loci were repeatedly detected across the two examined years (2020 and 2021), respectively. However, none was detected at the both sites. This could reflect the distinct photoperiods at these two sites (with their different sowing dates as noted in our response above).

6. In the field conditions, the determinate, semi-determinate or indeterminate may not easily to be discriminated due to the genetic epistasis or environmental effects. In legumes (pea, soybean, mung bean), I believe it is same in rice bean, *TFL1* determines the node termination and node numbers to control plant height. Plant height can be separated by node number (controlled by *TFL1*) and internode length (normally controlled by *GA*). I wonder whether the node numbers of 440 landraces had been investigated and GWAS should be analyzed which is more precisely to represent the effect of *TFL1*.

Response: Thanks for your helpful guidance. We do not have the node number data, but we have updated the discussion in the revised manuscript to include mention of this idea (Lines 441-444):

“Our GWAS analyses helped decipher the genetic basis of stem determinacy in rice bean, detecting that stem determinacy of rice bean is influenced by the *TFL1* gene; this gene has been implicated in determining the node termination and node numbers to control plant height and stem determinacy in many legume species^{26, 100, 101, 102}”

7. In relation of question 6, I suggest to divide population in Fig5 e, f into 2 nt deletion and non deletion of *TFL1*, other than Det and Indet. Because the phenotyping of Det and Indet may not accurate.

Response: Thanks for your suggestion. We have now divided the landraces from the SC group into landraces carrying homozygous reference alleles (designated as ‘Ref’) and those carrying the homozygous mutation (2-nt deletion) alleles (‘Del’), and have now conducted phenotypic comparisons (Fig. 5e in the revised manuscript) and estimation of divergence times between the two groups (Fig. 5f in the revised manuscript).

Fig. 5 | The molecular basis and selection history of stem determinacy in cultivated rice bean. InDel-based GWAS result from analysis of data for stem determinacy measured at the Nanning site in 2020. The peak InDel is indicated by the red circle. **b**, A 2-bp causative deletion (the peak InDel) introduced a premature termination codon in the first exon of the *TFL1* gene. Ref, reference; Del, deletion. **c**, The frequency distributions of three types of stem growth habit (indeterminate (Indet), semi-determinate (Semi-det), and determinate (Det)) among three groups comprising landraces carrying the homologous reference alleles (designated as ‘Ref’), heterozygous (‘Ref/Del’) or homologous mutation (2-bp deletion) alleles (‘Del’), respectively. Fisher’s exact tests were used to assess the significance of the differences for the proportion of the determinate type of stem growth habit between landraces carrying Ref and Del

alleles, and between landraces carrying Ref/Del and Del alleles. **d**, The geographical distributions of the 32 landraces carrying Del alleles from Southern-Central China. HB, Hubei province; HN, Hunan province; CQ, Chongqing province; GZ, Guizhou province; GX, Guangxi province. **e**, There was a significant improvement for the 32 landraces carrying Del alleles compared with the landraces carrying Ref alleles in the SC group for multiple human-desired traits including flowering time (DAE, days after emergence), branch number, seed length, and HGW (hundred seed weight). Significance was tested with Wilcoxon tests. **f**, Divergence time of the 32 landraces carrying Del alleles with the landraces carrying Ref alleles in the SC group, inferred using the SMC++ program⁶⁷, under a mutation rate $\mu = 1.5 \times 10^{-8}$ per site per generation¹⁴¹, and a generation time of one year.

8. It is very interesting to identify that the copy number of CYP78A6 is related to yield trait of rice bean. CNV always causes the dosage effect of the gene duplicated which can be reflected by transcriptional levels. Therefore, the expression levels of random selected landraces, for example to check the expressions of 20 landraces with one copy and 20 landraces with two copies, and compare the transcriptional differences between two groups.

Response: Following this suggestion, we planted all 440 landraces in the field in Beijing at June 15 2022 and selected 20 landraces with one copy and 20 landraces with two copies; the first fully expanded trifoliolate leaves were sampled at 14 days after sowing. qPCR analysis of the *VumCYP78A6* gene showed a significantly ($P < 0.01$) different expression level between these two groups, with 0.69 for the two-copies group vs. 0.34 for the one copy group on average. We have added these results in our revised manuscript as follows (Lines 378-383):

“The impact of this CNV on the expression of *VumCYP78A6* was also verified in a larger panel comprising 20 landraces with one copy and 20 landraces with two copies. Specifically, qPCR analysis of the *VumCYP78A6* gene for the first fully expanded trifoliolate leaves at 14 days after sowing showed a significantly ($P < 0.01$) higher expression level (~ 2-fold) in the 20 landraces with two copies than that in 20 landraces with one copy (Supplementary Fig. 24).”

Supplementary Fig. 24. qPCR analysis of seedlings (14 days after sowing) from 20 landraces with one copy of *VumCYP78A6* and 20 landraces with two copies. In the violin plots, central line: median values; bounds of box: 25th and 75th percentiles; whiskers: $1.5 \times \text{IQR}$ (IQR: the interquartile range between the 25th and 75th percentile). The red dot represents the average relative expression value.

We would again like to express our gratitude to the reviewer for the helpful guidance about how to improve our study; many thanks!

Reviewer #2 (Remarks to the Author):

The authors present the reference-quality genome assembly of *Vigna umbellata*, an underutilized legume crop from Asia. I agree with the authors that this represents a relevant resource that will facilitate accelerated and more targeted breeding of rice bean, with some potential in local diet. The genome assembly is of good quality and contiguity, with the relevant parameters (BUSCO, LAI, N50 etc) in an appropriate range. There seems to be a portion of sequence missing from the assembly, as compared to the estimated genome size (and also visible in the BUSCO scores). It would be helpful for the reader to learn whether this sequence is potentially located/missing in the centromeric regions. How well are the centromeres represented/resembled in this assembly?

Response: Thanks for the supportive review and helpful comments. We would like to again that the completeness of our rice bean genome assembly is the second highest (90.49%) for a for *Vigna* species but has the highest contiguity (N50 = 18.26 Mb) among the published genome assemblies for this genus (we have prepared a Response document table “Table R1” that presents this information). Our final genome assembly has 351 contigs; we mistakenly reported this as 592 in the originally submitted manuscript (to clarify, 592 was the number of contigs assembled using Pacbio long reads and CANU software (Koren et al., 2017). After further assembly based on these 592 CANU contigs and corrected Pacbio long reads using HERA algorithm (Du et al., 2019), the final assembly was consisted of 351 contigs. The number of contigs in the final assembly has been corrected in the Lines 100 and 505, and Table 1 of the revised manuscript.

We examined the 100 bp flanking region of all the 87 assembly gaps in the final assembly, and the results revealed that 78 gaps (89.66%) have more than one flanking region with high proportion (> 90%) of repeat sequences, suggesting that most of gaps were caused by the incomplete assembly of the repeat sequences. Centromeres consists of highly repetitive DNA sequences and are considered as the most difficulty genome regions to assemble (Nurk et al., 2022). To evaluate the completeness of centromeres in our genome assembly, we have now used a published method (Su et al., 2021) to identify centromere-specific repeats. We found a 217-bp centromere-specific repeat (Supplementary Fig. 28) and used this

to predict candidate centromere regions for each of the 11 chromosomes based on the distribution of this repeat (Supplementary Table 11). Unfortunately, all the 11 candidate centromere regions (with lengths from 0.29 Mb to 3.07 Mb) contained more than one assembly gap, suggesting none of the centromere sequences was fully assembled; future efforts using long sequencing reads (likely the ultra-long ONT reads) should help to ‘close these gaps’. We have added this new information in the ‘Discussion’ section of the revised manuscript (Lines 400-407):

“Our rice bean genome assembly still contains 87 gaps, 78 of which have more than one flanking region (100 bp) with a high proportion (> 90%) of repeat sequences, suggesting that most of gaps were caused by the incomplete assembly of the repeat sequences, similar to previous reports^{75,76}. We also predicted the candidate centromere regions using a previously published method⁷⁷ (See methods) and found that all 11 of the candidate centromere regions contained more than one assembly gap, suggesting that none of the centromere sequences was fully assembled (Supplementary Table 11); future efforts using long sequencing reads (likely the ultra-long ONT reads) should help to ‘close these gaps’”.

We have also added content to the ‘Methods’ section of the revised manuscript is as follows (Lines 519-524):

“We analyzed the density distribution of the top-50 most abundant repeat subfamilies in 100 Kb windows (using RepeatMasker), and used BEDtools¹¹³ to merge the results with the parameter ‘-d 100000’. The rnd-6_family-604 subfamily (a 217-bp repeat) was identified as a centromere-specific repeat (Supplementary Fig. 28). The candidate centromere regions were also predicted according to the density distribution of this centromere-specific repeat (Supplementary Table 11).”

Table R1. The completeness of the published genome assemblies for *Vigna* species

Species	Estimated genome size by kmer analysis (Mb)	Genome assembly size (Mb)	Completeness (%)	Contig N50 (Mb)	References
V. umbellata	525.6	475.64	90.49	18.26	Our study
V. unguiculata	560.3	519	92.63	10.91	Lonardi et al., 2019

V. stipulacea	445.1	387.7	87.10	1.94	Takahashi et al., 2019
V. mungo	574	498.9	86.92	5.24	Pootakham et al., 2021
V. angularis	542	466.7	86.11	1.52	Yang et al., 2015
V. radiata	543	431	79.37	0.04	Kang et al., 2014

Supplementary Fig. 28. The frequency distribution of the 217-bp centromere-specific repeat.

Supplementary Table 11. The position and lengths of the potential centromere.

Chr	Length (bp)	Centromere			
		Start (bp)	End (bp)	Length (Mb)	Number of gaps
1	65,796,979	33,160,000	34,360,000	1.20	2
2	47,516,185	21,420,000	24,330,000	2.91	3
3	40,359,536	17,800,000	18,480,000	0.68	1
4	54,125,813	17,210,000	19,520,000	2.31	2
5	39,577,938	22,350,000	23,060,000	0.71	5
6	38,145,303	14,830,000	17,900,000	3.07	2
7	31,513,027	15,040,000	15,710,000	0.67	1
8	46,389,798	27,740,000	28,290,000	0.55	1
9	34,248,148	16,950,000	18,780,000	1.83	1
10	29,464,087	13,510,000	15,910,000	2.40	1
11	38,052,607	21,040,000	21,330,000	0.29	1

References:

- Du, H. & Liang, C. Assembly of chromosome-scale contigs by efficiently resolving repetitive sequences with long reads. *Nat. Commun.* **10**, 5360 (2019).
- Kang, Y. et al. Genome sequence of mungbean and insights into evolution within *Vigna* species. *Nat.*

Commun. **5**, 5443 (2014).

Koren, S. et al. Canu: scalable and accurate long-read assembly via adaptive *k*-mer weighting and repeat separation. *Genome Res.* **27**, 722-736 (2017).

Lonardi, S. et al. The genome of cowpea (*Vigna unguiculata* [L.] Walp.). *Plant J.* **98**, 767-782 (2019).

Nurk, S. et al. The complete sequence of a human genome. *Science* **376**, 44-53 (2022).

Pootakham, W. et al. A chromosome-scale assembly of the black gram (*Vigna mungo*) genome. *Mol. Ecol. Resour.* **21**, 238-250 (2021).

Su, X. et al. A high-continuity and annotated tomato reference genome. *BMC Genomics.* **22**, 898 (2021).

Takahashi, Y. et al. Domesticating *Vigna stipulacea*: a potential legume crop with broad resistance to biotic stresses. *Front. Plant Sci.* **10**, 1607 (2019).

Yang, K. et al. Genome sequencing of adzuki bean (*Vigna angularis*) provides insight into high starch and low fat accumulation and domestication. *Proc. Natl Acad. Sci USA* **112**, 13213-13218 (2015).

The collinear genes/segments displayed in the Circos illustration (Figure 1b; track VIII) suggest that there is quite a large number of those...how does this go together with the smaller % of duplicated genes in the BUSCO evaluation and the conclusion that no recent WGD was detected? What were the criteria for collinearity in Figure 1b? Further, in line 148, how many “syntenic blocks” and homologous gene pairs were identified, and how?

Response: Thanks for your comment. After aligning all the 26,736 protein sequences of our rice bean genome to itself using BLASTP (-outfmt 6 -evalue 1e-10), we used the popular program MCScanX (Wang et al., 2012) with the default parameters based on the BLASTP output to identify the syntenic blocks and homologous gene pairs within the rice bean genome (as used in studies for plants such as tepary bean, soybean, and mungbean (Kang et al., 2014; Moghaddam et al., 2021; Wang et al., 2021)). A total of 332 syntenic blocks were identified, including 8,052 homologous genes accounting for ~30.12% of all genes. This proportion is similar with that of species without a recent WGD event like *V. radiata* (24.93%) and *P. vulgaris* (25.38%) (Wang et al., 2017), and much lower than that of species with recent WGD events like soybean (~54%) (Liu et al., 2020). The density distribution of Ks values of these homologous gene pairs only has two peaks formed by two well-known ancient WGD events before

speciation—LCT (Schmutz et al., 2010) and ECH (Bowers et al., 2003)—which was also reported for other *Vigna* species (Wang et al., 2017; Pootakham et al., 2021). Our data thus suggests that there have been no recent WGD events after speciation in the rice bean genome. To avoid confusion, we have added the number of syntenic blocks and homologous gene pairs in the revised manuscript as follows (Lines 151-154):

“To investigate WGD events in rice bean, we identified 332 syntenic blocks within its genome (including 8,052 homologous genes accounting for ~ 30.12% of all genes) (Fig. 1B) and estimated synonymous nucleotide substitutions at synonymous sites (Ks) for homologs.”

References:

- Bowers, J., Chapman, B., Rong, J. & Paterson, A. Unravelling angiosperm genome evolution by phylogenetic analysis of chromosomal duplication events. *Nature* **422**, 433–438 (2003).
- Kang, Y. et al. Genome sequence of mungbean and insights into evolution within *Vigna* species. *Nat. Commun.* **5**, 5443 (2014).
- Liu, Y. et al. Pan-Genome of Wild and Cultivated Soybeans. *Cell* **182**, 162-176.e13 (2020).
- Moghaddam, S. et al. The tepary bean genome provides insight into evolution and domestication under heat stress. *Nat. Commun.* **12**, 2638 (2021).
- Pootakham, W. et al. A chromosome-scale assembly of the black gram (*Vigna mungo*) genome. *Mol. Ecol. Resour.* **21**, 238-250 (2021).
- Schmutz, J. et al. Genome sequence of the palaeopolyploid soybean. *Nature* **463**, 178–183 (2010).
- Schmutz, J. et al. A reference genome for common bean and genome-wide analysis of dual domestications. *Nat. Genet.* **46**, 707-713 (2014).
- Wang, L. et al. Altered chromatin architecture and gene expression during polyploidization and domestication of soybean. *Plant Cell* **33**, 1430-1446 (2021).
- Wang, J. et al. Hierarchically Aligning 10 legume genomes establishes a family-level genomics platform. *Plant Physiol.* **174**, 284-300 (2017).
- Wang, Y. et al. MCScanX: a toolkit for detection and evolutionary analysis of gene synteny and collinearity. *Nucleic Acids Res.* **40**, e49 (2012).

Figure 3b: what is the beige/brown color? Background?

Response: In the STRUCTURE bar plot (Fig. 3b), the length of each colored component corresponds to the proportion of each inferred ancestry for a length-fixed bar. When $K = 4$ in the Fig. 3b, the bars with predominant proportion of beige color represent those determinate-type landraces with a 2-nt deletion mutation that causes premature termination of translation of the *TFL1* transcript. Note that these landraces were collected from a relatively narrow geographical region (South-Central China comprising five adjacent provinces: Chongqing, Hunan, Hubei, Guizhou, and Guangxi) and had close genetic relationship as inferred by a phylogenetic tree (Fig. 3b). Moreover, these all have human-desired traits and there is clear genetic divergence between these and other landraces within the SC group (Lines 339-341 of the revised manuscript). To avoid confusion, we have rephrased the sentence in the revised manuscript as (Lines 335-338):

“We also observed that these landraces with 2-nt deletion (represented by the bars with predominant proportion of beige color in the Supplementary Fig. 18) were genetically distinct from other landraces within the SC group using model-based clustering ($K = 4$), an inference that was further supported by a moderate level of differentiation ($F_{ST} = 0.11$).”

I do have some concerns/questions with the observed flowering time variation that the authors relate to latitude. These data are from different years, without any replicates or controls as far as I understand. How sure can you be that the observed differences are not simply due to climate/temperature differences or other environmental factors in the years/locations? I understand/assume all 440 landraces been grown in all three locations?

The GWAS identified candidate genes appear plausible to me, and the conclusions for latitude and allele analyses is interesting, but I would like to see some kind of control for the flowering time variation.

Response: Regarding this comment specifically, to be clear, we need to stress that we do have multi-year evidence for all of the associations that we present in our study. Specifically, we have two-year (2020 and 2021) phenotyping data for traits including flowering time at all three sites, for stem

determinacy at the Nanning site, and for both HGW and seed length at the Sanya and Nanning sites. GWAS of this phenotype data revealed 15 QTLs that were repeatedly detected across two years (containing candidate genes including *FUL*, *FT*, *PRR3*, *TFL1*, and *VumCYP78A6*) (Supplementary Table 9). Note that we have added Supplementary Figures and a Table to present the Manhattan plots of GWAS for flowering-time data (Supplementary Fig. 8), seed yield traits (Supplementary Fig. 20), stem determinacy (Supplementary Fig. 16), as well as the detailed information for all associated loci (Supplementary Data 5).

Kindly note that in the originally submitted manuscript, we did have claims of associations for three traits that lacked multi-year evidence (plant height at the Nanning site in 2020; pod length and pod width at the Nanning site in 2020 and the Sanya site in 2021). These have been dropped from the revised manuscript.

Supplementary Table 9. GWAS QTLs detected across two years (2020 and 2021)

Trait	Site	QTL	Years ^a	PVE (%)	Candidate gene
Flowering time	Sanya	Chr8:1,909,300-1,919,793	2	10.57	-
		Chr11:6,142,933-6,162,249	2	14.86	Vum_11G00418 (FUL)
Stem determinacy	Nanning	Chr4:35,931,101-35,996,258	2	8.23	Vum_04G01668 (FT)
	Beijing	Chr2:38,647,190-38,647,190	2	17.30	Vum_02G01965 (PRR3)
	Nanning	Chr3:12,535,769-16,852,788	2	8.69	-
		Chr4:42,022,544-54,020,700	2	41.43	Vum_04G02513 (TFL1)
		Chr5:13,477,451-13,789,216	2	32.19	-
		Chr6:2,759,250-6,161,777	2	12.42	-
		Chr7:18,729,386-20,781,885	2	8.02	-
		Chr9:28,394,569-28,697,545	2	8.24	-
HGW	Sanya	Chr9:29,081,975-29,126,729	2	8.88	Vum_09G01129 & Vum_09G01130
		Chr9:29,030,437-29,126,729	2	15.74	
Seed length	Sanya	Chr9:29,030,437-29,126,729	2	11.39	(VumCYP78A6-1 & VumCYP78A6-2)
	Nanning	Chr9:29,030,437-29,126,729	2	16.17	

^a Number of years in which the QTL was detected.

Supplementary Fig. 8. Manhattan plots of GWAS for flowering-time data measured at the Sanya (a), Nanning (b), and Beijing (c) sites across two years (2020 and 2021).

Supplementary Fig. 20. Manhattan plots of GWAS for seed yield traits measured at the Nanning (a and b) and Sanya (c and d) sites across two years (2020 and 2021). HSW, hundred seed weight; SL, seed length.

Supplementary Fig. 16. Manhattan plots of SNP-based (a) and InDel-based (b) GWAS for stem determinacy measured at the Nanning site across two years (2020 and 2021). The peak InDel (2-nt deletion) is indicated by the red circle.

Another major concern is with data availability: I could not access any (!) of the given accession codes or figshare download links. I understand authors may want to wait until publication but should provide a possibility for data download and inspection during reviewing. As a result I cannot comment on data quality, completeness and presentation.

Response: The available links for the raw sequencing data deposited at the Sequence Read Archive of the National Center for Biotechnology Information (NCBI) under BioProjects PRJNA819955 and PRJNA803965 are as follows:

PRJNA819955:

<https://dataview.ncbi.nlm.nih.gov/object/PRJNA819955?reviewer=u6gomm43s5frlspkhp5df22tle>

PRJNA803965:

<https://dataview.ncbi.nlm.nih.gov/object/PRJNA803965?reviewer=c25h5otlhifvghjm9j6a5t9rbj>

The available links for the genome assembly sequence and the gene annotation file deposited in the Figshare database are as follows:

<https://figshare.com/s/f395ccba2e7459116249>.

We have tested the availability status of these links: all were accessible from China on 15 August 2022.

Kindly note that the raw transcriptome data for pod tissue (under BioProject PRJNA820123) has been dropped from the revised manuscript because all the pod traits that lacked multi-year evidence have been removed.

We would like to take this opportunity to again thank the reviewer for the helpful guidance.

Reviewer #3 (Remarks to the Author):

The authors report the high-quality genome of *Vigna umbellata* and the analysis of germplasm population. The approach and methods analysis are gold standard and resulted in outstanding results. The authors revealed important genes associated with agronomic traits from GWAS analysis of the germplasm. There are only minor points to be added to make the manuscript even stronger.

1. Gene synteny block analysis between *Vigna* species would demonstrate how genome structure and order of gene are conserved in this genus.

Response: Thanks for your helpful suggestion. We have now completed a new gene synteny block analysis between rice bean and its closely related species in *Vigna* genus, and added the new content in the revised manuscript as follows (Lines 140-144):

“This view was also supported by a gene synteny analysis between rice bean and its closely related species in the *Vigna* genus based on protein sequences using the MCScanX program⁴⁴, which revealed that (as expected) rice bean had higher conservation with *V. angularis* in terms of gene structure and order as compared to other *Vigna* species (Supplementary Fig. 4; Supplementary Table 7).”

We also added content in the ‘Methods’ section as follows (Lines 565-566):

“The synteny figure was plotted using the NGenomeSyn program (<https://github.com/hewm2008/NGenomeSyn>).”

Supplementary Fig. 4. Syntenic genes between *Vigna umbellata* (Vum) and its closely related *Vigna* species (*V. unguiculata* (Vun), *V. angularis* (Van), *V. radiata* (Vra), and *V. stipulacea* (Vst)). Note that the Vst genome only shows the top 47 longest scaffolds (> 1 Mb).

Supplementary Table 7. Summary statistics of syntenic blocks and genes between the Vum genome and its closely related *Vigna* species

Species in Vigna genus	Van	Vst	Vra	Vun
Syntenic block number	639	714	831	803
Number and percentage of syntenic genes in Vum genome	20,985/79.40%	19,459/73.62%	17,980/68.03%	19,990/75.63%

V. angularis, Van; *V. stipulacea*, Vst; *V. radiata*, Vra; *V. unguiculata*, Vun

2. Selection (Ka/Ks) between tropical and temperate legumes would be interesting: *V. umbellata* is widespread across both tropical and temperate regions.

Response: Thanks for your helpful guidance; to our understanding, given that the Ks/Ks ratio is more suitable for the comparison between two genomes (Nekrutenko et al., 2002; Zeng et al., 2014; Wang et al., 2104), it would be more informative to conduct genomic selection analysis for the population-level genomic data using F_{ST} measurement (Lam et al., 2019; Maccaferri et al., 2019; Yue et al., 2022; Hu et al., 2022) Thus, we have now completed a selective sweep analysis for the three comparisons (SSA vs. SC, SSA vs. NC, and SC vs. NC) using F_{ST} (Fig. 3d). The SSA group contains landraces mostly from tropical regions (South and Southeast Asia); the SC and NC groups respectively contain landraces mostly

from the South and North of the temperature region in China. We have added these new contents in the revised manuscript as follows (Lines 218-230):

“We searched for putatively selective regions with outliers (top 5%) of F_{ST} over 20-kb windows for the three comparisons (SSA vs. SC, SSA vs. NC, and SC vs. NC). We detected 473, 512, and 444 outlier regions for these three comparisons, respectively occupying 5.67% (26.95 Mb), 5.92% (28.15Mb), and 5.59% (26.57 Mb) of the genome and including 1,894, 1,950, and 1,296 protein-coding genes (Supplementary Data 4). A MapMan analysis of all the selected genes indicated that these genes were significantly enriched for annotations related to biological processes such as “phytohormone action”, “nutrient uptake”, and “circadian clock system” (Supplementary Fig. 6). Notably, among the genes related to “circadian clock system”, we found four orthologs of reported flowering time genes in *A. thaliana* using FLOR-ID (FLOWeRing Interactive Database⁵⁸), including *TOCI*⁵⁹, *PRR3*⁶⁰, and two *LHY1*⁶¹ genes between the SSC and SC groups, of which one *LHY1*⁶¹ apparently also underwent selection between the SSC and NC groups (Supplementary Fig. 7). These flowering time genes could plausibly have contributed to the adaptation of rice bean landraces to different latitudes.”

Supplementary Fig. 6. MapMan enrichment analysis of all the selected genes for the three comparisons (SSA vs. SC, SSA vs. NC, and SC vs. NC).

Supplementary Fig. 7. Genomic divergence signals for the three comparisons (SSA vs. SC, SSA vs. NC, and SC vs. NC) and homologous genes related to “circadian clock system” within these signals. The red lines represent the diverged genomic signals and the black dashed lines indicate the locations of the homologous genes related to “circadian clock system”.

References:

- Hu, G. et al. Two divergent haplotypes from a highly heterozygous lychee genome suggest independent domestication events for early and late-maturing cultivars. *Nat. Genet.* **54**, 73-83 (2022).
- Lam, M. et al. Comparative genetic architectures of schizophrenia in East Asian and European populations. *Nat. Genet.* **51**, 1670-1678 (2019).
- Maccaferri, M. et al. Durum wheat genome highlights past domestication signatures and future improvement targets. *Nat. Genet.* **51**, 885-895 (2019).
- Nekrutenko, A., Makova, K. & Li, W. et al. The K_A/K_S ratio test for assessing the protein-coding potential of genomic regions: an empirical and simulation study. *Genome Res.* **12**, 198-202 (2002).
- Wang, W. et al. Cassava genome from a wild ancestor to cultivated varieties. *Nat. Commun.* **5**, 5110 (2014).
- Yue, J. et al. SunUp and Sunset genomes revealed impact of particle bombardment mediated transformation and domestication history in papaya. *Nat. Genet.* **54**, 715-724 (2022).
- Zeng, X. et al. The draft genome of Tibetan hulless barley reveals adaptive patterns to the high stressful Tibetan Plateau. *Proc. Natl. Acad. Sci. USA* **112**, 1095-1100 (2015).

3. Flowering time: Please describe photoperiod of the three locations (Sanya, Naning, Beijing) during the time of experiment performed, describe action mode whether these alleles (FUL, FT, PRR3) are dominant, recessive or co-dominant.

Response: Thanks for your helpful suggestions. We have added the descriptions of the photoperiod at the three sites during the time of experiment in the ‘Methods’ section of the revised manuscript as follows (Lines 480-485):

“Supplementary Fig. 27 presents the day length (per day) during the ~ five month growth period for the three sites. The day length differs obviously among the three sites (but was very similar between the two observation years). The average day lengths of Beijing during the first four months (during which all the landraces opened the first flower) was the longest (13.94 and 13.93 h) in both 2020 and 2021, followed by the Nanning site (12.52 and 12.53 h) and the Sanya site (11.28 and 11.28 h).”

For flowering-time related loci (*FUL*, *FT*, and *PRR3*), we used the R package “SNPassoc” (González et al., 2007) to perform association analysis of the alleles based on several genetic models (co-dominant, dominant, recessive, over-dominant, or additive). The inheritance model with the best fit (that is, with the lowest AIC [akaike information criteria] value) for alleles of *FUL*, *FT*, and *PRR3* loci is additive, dominant, and additive, respectively (Supplementary Table 10). We have added these results in the revised manuscript as follows (Lines 285-288):

“We also inferred the model of inheritance for these alleles and found that the best models for *FUL*, *FT*, and *PRR3* loci were additive, dominant, and additive, respectively (Supplementary Table 10; See methods).”

We have also added new content in the ‘Methods’ section of the revised manuscripts follows (Lines 617-622):

“To infer the most likely inheritance model of alleles for the flowering-time related loci (*FUL*, *FT*, and *PRR3*), we used the R package “SNPassoc”¹⁴⁶ to perform association analysis of the alleles based on several genetic models (co-dominant, dominant, recessive, over-dominant, or additive). The model with smallest AIC (akaike information criteria) value was identified as the best fitting genetic model.”

Supplementary Fig. 27. The day length (hour) per day during the growth period at Sanya site (18° N) in 2019-2020 and 2020-2021, at Nanning site (22° N) in 2020 and 2021, and at Beijing site (40° N) in 2020 and 2021.

Supplementary Table 10. AIC (akaike information criteria) value of five different genetic models used in the association analysis of the alleles associated with flowering-time data measured in two years at the three distinct sites.

Alleles	Year	AIC				
		Co-dominant	Dominant	Recessive	Over-dominant	Additive
FUL	2020	-147.20	-136.30	-120.10	-35.70	-149.06
	2021	-169.10	-166.50	-146.80	-92.10	-169.82
FT	2020	-8.00	-9.50	32.10	78.00	-2.32
	2021	52.10	50.20	93.40	116.70	61.50
PRR3	2020	-351.00	-347.70	-340.20	-269.10	-352.80
	2021	-356.00	-352.20	-340.50	-272.50	-357.48

References:

González, J. et al. SNPassoc: an R package to perform whole genome association studies. *Bioinformatics* **23**, 654-655 (2007).

4.Fig2A: misspelling of Vigra

Response: Thanks for your comment. We have corrected this mistake in the Fig. 2A of our revised manuscript and have checked this throughout the text.

We would like to take this opportunity to thank the reviewer for the supportive comments and the excellent guidance about how to improve our study.

Reviewers' Comments:

Reviewer #1:

Remarks to the Author:

The authors had addressed all the concerns and I have no further comments.

Reviewer #2:

Remarks to the Author:

Most of my concerns were addressed by the authors in the revised version of the manuscript.

While I appreciate the corrections applied to the phenotyping associations, I still wonder whether the conclusions drawn for the traits (stem determinacy and seed length) analysed in multi-year studies only in one or two sites are robust enough, given that the authors try to associate these to latitude. I acknowledge that interesting QTLs were detected in this analysis setup but I think that potential pitfalls and limitations with this approach need at least be discussed.

Reviewer #3:

Remarks to the Author:

No further comments.

Response to Reviewers' comments

Reviewer #1 (Remarks to the Author):

The authors had addressed all the concerns and I have no further comments.

Response:

We would first like to thank the Reviewer for the supportive review and for the very helpful guidance about how to improve our study.

Reviewer #2 (Remarks to the Author):

Most of my concerns were addressed by the authors in the revised version of the manuscript.

While I appreciate the corrections applied to the phenotyping associations, I still wonder whether the conclusions drawn for the traits (stem determinacy and seed length) analysed in multi-year studies only in one or two sites are robust enough, given that the authors try to associate these to latitude. I acknowledge that interesting QTLs were detected in this analysis setup but I think that potential pitfalls and limitations with this approach need at least be discussed.

Response:

We appreciate the reviewer's support and are thankful for the guidance about our study.

We have updated the discussion in the revised manuscript to include mention of this idea (Lines 451-454):

“Although QTLs for stem determinacy and seed yield-related traits were detected by our GWAS analyses in one and two environments respectively, further efforts should be made to investigate the robustness of these QTLs in more different environments.”

We would again like to express our gratitude to the reviewer for the helpful guidance about how to improve our study; many thanks!

Reviewer #3 (Remarks to the Author):

No further comments.

Response:

We thank the reviewer for the guidance and care in helping us to improve our manuscript.